# Solving Diffusion ODEs with Optimal Boundary Conditions for Better Image Super-Resolution

**Yiyang Ma[1], Huan Yang[2], Wenhan Yang[3], Jianlong Fu[2], Jiaying Liu[1]***

[1]Wangxuan Institute of Computer Technology, Peking University, [2]Microsoft Research,
[3]Pengcheng Laboratory
[1]{myy12769, liujiaying}@pku.edu.cn,
[2]{huayan, jianf}@microsoft.com, [3]yangwh@pcl.ac.cn

## Abstract

Diffusion models, as a kind of powerful generative model, have given impressive results on the image super-resolution (SR) tasks. However, due to the randomness introduced in the reverse process of diffusion models, the performances of diffusion-based SR models fluctuate at every time of sampling, especially for the samplers with few resampled steps. This inherent randomness of diffusion models results in ineffectiveness and instability, making it challenging for users to guarantee the quality of SR results. However, our work takes this randomness as an opportunity: fully analyzing and leveraging it to the construction of an effective plug-and-play sampling method that has the potential to benefit a series of diffusion-based SR methods. More in detail, we propose to steadily sample high-quality SR images from pre-trained diffusion-based SR models by solving diffusion ordinary differential equations (*diffusion ODE*s) with optimal boundary conditions (BCs) and analyze the characteristics between the choices of BCs and their corresponding SR results. Our analysis shows the route to obtain an approximately optimal BC via an efficient exploration in the whole space. The quality of SR results sampled by the proposed method with fewer steps outperforms the quality of results sampled by current methods with randomness from the same pre-trained diffusion-based SR model, which means that our sampling method "boosts" current diffusion-based SR models without any additional training.

## 1 Introduction

Diffusion models (Ho et al., 2020) have drawn great research attention within the domain of computer vision because of their great capacity for image generation. Therefore, it is intuitive to leverage such powerful models to tackle the demanding task of image super-resolution (SR). The diffusion-based image SR task is modeled as generating high-quality images by diffusion models conditioned on corresponding low-resolution images (Saharia et al., 2022c; Li et al., 2022; Shang et al., 2023; Sahak et al., 2023). However, the reverse process (*i.e.*, generating process) of diffusion models, including randomness (Ho et al., 2020; Song et al., 2021a;b), leads to unstable performances of the diffusion-based SR methods. In other words, the users cannot guarantee the quality of SR results if they lack a principled approach and can only rely on random sampling from diffusion-based models. The previous methods did not consider or explore the issue of randomness. Although multiple random samplings methods can lead to reasonable SR images using well-trained diffusion-based SR models. However, we cannot guarantee the quality of one-time sampling, and the sampled results on average still fall short of optimal quality, with significant performance gaps. Thus, it is critical to pursue a stable sampling method that generates SR images from pre-trained diffusion models with guaranteed good performances.

Most current diffusion-based SR works (Saharia et al., 2022c; Li et al., 2022; Shang et al., 2023; Sahak et al., 2023; Wang et al., 2023) focus on the model design instead of the sampling method. The

---

*Corresponding author.

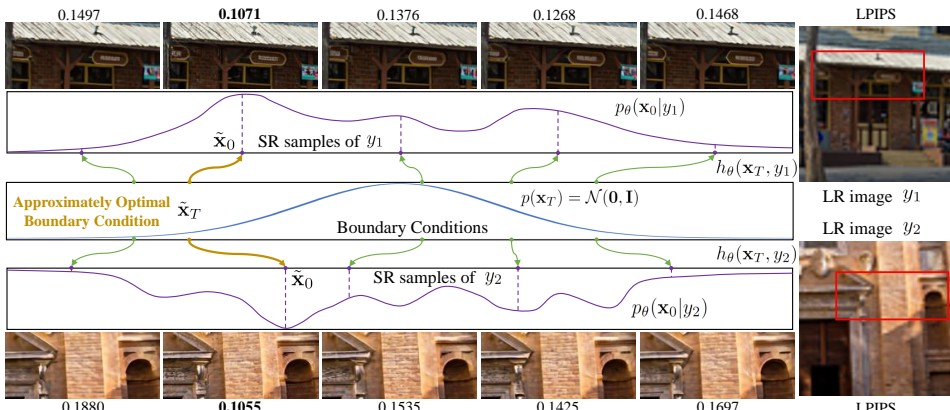

Figure 1:    Given a well-trained diffusion-based SR model, by solving *diffusion ODE*s, we can sample reasonable SR results with different BCs $\mathbf{x}_T$ as the figure shows. However, there is instability in the performances of each BC $\mathbf{x}_T$. We manage to find an approximately optimal BC $\tilde{\mathbf{x}}_T$ which can be projected to the sample $\tilde{\mathbf{x}}_0$ with nearly the highest probability density by the solution $h_\theta(\tilde{\mathbf{x}}_T, \mathbf{y})$ to *diffusion ODE*. Based on our analysis in the Sec. 3.2, $\tilde{\mathbf{x}}_T$ is shared by different LR images $\mathbf{y}_i$. The method of finding $\tilde{\mathbf{x}}_T$ refers to the Sec. 3.3 **[Zoom in for best view]**

most commonly used sampling method for diffusion-based SR works is a resampled DDPM sampler with 100 steps (DDPM-100) instead of the original DDPM sampler with 1000 steps of the training noise schedule (DDPM-1000), due to its significantly reduced time cost, despite the trade-off in SR image quality. It is first introduced by SR3 (Saharia et al., 2022c) from WaveGrad (Chen et al., 2021). Later works following SR3 use DDPM-100 as a default setting. These discrete-time DDPM samplers sample from a Gaussian distribution with learned parameters at each step, resulting in instability. Song et al. (2021b) demonstrate that such discrete-time DDPM samplers can be regarded as solving diffusion stochastic differential equations (*diffusion SDE*s) and further give ordinary differential equations which share the same marginal probability densities as *diffusion SDE*s. Such ordinary differential equations are referred to as *diffusion ODE*s. Different from *diffusion SDE*s, given a boundary condition (BC) $\mathbf{x}_T$, one can solve the *diffusion ODE*s via ODE samplers (*e.g.*, DDIM (Song et al., 2021a), DPM Solver (Lu et al., 2022)) to get an exact solution $\mathbf{x}_0$. Nevertheless, the BCs $\mathbf{x}_T \sim \mathcal{N}(\mathbf{0}, \mathbf{I})$ also come with randomness and lead to the instability issue in sampling SR images. Hence, it is highly desirable to obtain a principled way for estimating the optimal BC $\mathbf{x}_T^*$ to steadily offer sampled SR images with high quality.

In this paper, we analyze the characteristics of the optimal BC $\mathbf{x}_T^*$ of *diffusion ODE*s of SR models and propose an approach to approximate the optimal BC $\tilde{\mathbf{x}}_T$ by exploring the whole space with the criterion of a reference set containing $R$ HR-LR image pairs $\mathcal{R} = \{(\mathbf{z}_i, \mathbf{y}_i)\}_{i=1}^R$ which is a small subset of the training dataset. Then, we can steadily generate high-quality SR images by solving the *diffusion ODE*s of the trained diffusion-based SR model with the above derived approximately optimal BC $\tilde{\mathbf{x}}_T$. We establish that the optimal boundary condition $\mathbf{x}_T^*$ utilized to solve the *diffusion ODE*s in diffusion-based SR models is independent of the LR image inputs. Thus, we only need to prepare the approximately optimal BC $\tilde{\mathbf{x}}_T$ once to sample SR images of other unseen LR images. The experiment demonstrates that this simple independence assumption empirically offers impressive performance in a plug-and-play manner. The main idea is shown in Fig. 1.

We evaluate our method on both bicubic-SR and real-SR degradation settings. For bicubic-SR, we train a vanilla diffusion-based SR model which simply concatenates LR images with noisy images $\mathbf{x}_t$ as the architecture proposed in SR3 (Saharia et al., 2022c). For real-SR, we apply our method to StableSR (Wang et al., 2023), which finetunes pre-trained Stable Diffusion (Rombach et al., 2022) on real-SR data. Experiments show that the quality of SR images sampled by few-step *diffusion ODE* samplers with our explored BC $\tilde{\mathbf{x}}_T$ significantly outperforms the quality of results sampled by existing methods owning the same architecture. Our method is not restricted to any specific architecture of diffusion-based SR models. As the models we utilize in bicubic-SR and real-SR are quite different, the versatility of our method can be demonstrated by experiments. Therefore, any diffusion-based SR model can leverage the proposed method to steadily sample high-quality

SR images with only a few steps, and achieve improved performance. In this way, our method can "boost" existing diffusion-based SR models in the plug-and-play manner.

## 2 RELATED WORK

### 2.1 IMAGE SUPER-RESOLUTION

Image super-resolution has drawn great research interest in recent years (Dong et al., 2014; Kim et al., 2016; Tong et al., 2017; Lim et al., 2017; Ledig et al., 2017; Wang et al., 2018; Zhang et al., 2018b; Liang et al., 2021). As a pioneer work of deep-learning based SR methods, SRCNN (Dong et al., 2014) builds a 3-layer convolutional neural network to map LR patches to SR patches with the criterion of MSE between SR patches and HR patches, getting better PSNR than traditional methods. SRResNet (Ledig et al., 2017) introduces residual connections into SR networks, achieving impressive performances. RCAN (Zhang et al., 2018b) uses channel-attention mechanism to learn local-correlation which is crucial to the SR task. SWINIR (Liang et al., 2021) leverages vision transformers (Dosovitskiy et al., 2021; Liu et al., 2021) to build backbones of SR neural networks and outperforms CNN-based NNs.

However, PSNR between SR images and HR images has a gap with the visual quality of SR images. Using generative models can synthesize more perceptually pleasant results. Thus, SRGAN (Ledig et al., 2017) introduces GANs (Goodfellow et al., 2014) to SR tasks. Furthermore, Menon et al. (2020); Yang et al. (2021); Chan et al. (2021) incorporate pre-trained GANs from specific domains into their SR frameworks, leveraging the generative capabilities of these GANs. PixelSR (Dahl et al., 2017) uses auto-regressive models to generate SR images pixel-by-pixel. SRFlow (Lugmayr et al., 2020) models SR tasks by normalizing flow-based models (Kingma & Dhariwal, 2018). SR3 (Saharia et al., 2022c) first uses diffusion models (Ho et al., 2020; Song et al., 2021b) to generate SR images conditioned on corresponding LR images. DDRM (Kawar et al., 2022) designs a training-free algorithm to guide pre-trained diffusion models to generate high-quality images which are consistent with the LR images. StableSR leverages pre-trained Stable Diffusion (Rombach et al., 2022) as a generative prior.

### 2.2 DIFFUSION MODELS

In recent years, diffusion models (Ho et al., 2020; Song et al., 2021b), a type of generative model, have achieved impressive results across various research domains, including image generation (Dhariwal & Nichol, 2021; Nichol & Dhariwal, 2021), text-to-image generation (Nichol et al., 2022; Ramesh et al., 2022; Saharia et al., 2022b), multi-modal generation (Ruan et al., 2023; Ma et al., 2023) and so on. Diffusion models are first proposed by Sohl-Dickstein et al. (2015) and simplified as DDPM by Ho et al. (2020) which can be trained as several simple denoising models. ImprovedDDPM (Nichol & Dhariwal, 2021) proposes to learn the variance of each reverse step and AnalyticDPM (Bao et al., 2022) claims that such variances have analytic forms which not need to be learned. Song et al. (2021b) extend the diffusion models with discrete Markovian chains to continuous differential equations. Ho et al. (2022) propose to train diffusion models by "velocity", getting more efficiency. Rombach et al. (2022) build diffusion models on latent spaces instead of image spaces, reducing the training and inferring cost.

In terms of applying diffusion models, GLIDE (Nichol et al., 2022) first proposes to build a diffusion model to generate images from descriptive texts. DALL·E 2 (Ramesh et al., 2022) and Imagen (Saharia et al., 2022b) design better architecture and use more computing resources, achieving better performances. Palette (Saharia et al., 2022a) first applies diffusion models to image-to-image translation tasks. DreamBooth (Ruiz et al., 2023) finetunes pre-trained text-to-image diffusion models to achieve the goal of subject-driven generation. MM-Diffusion (Ruan et al., 2023) generates aligned audios and videos at the same time. Singer et al. (2023) create novel videos from texts without text-to-video data. These works prove that diffusion models have strong generative abilities.

## 3 SAMPLING SR IMAGES WITH OPTIMAL BCS OF *Diffusion ODEs*

We first review diffusion models and their continuous differential equations, then analyze the optimal BCs $\mathbf{x}_T^*$ used by *diffusion ODEs* to sample SR images from diffusion-based SR models, last depict the method of approximating the optimal BCs $\tilde{\mathbf{x}}_T$ in Eqn. 20 with the criterion of a reference set containing $R$ image pairs. With the approximately optimal $\tilde{\mathbf{x}}_T$, we can sample high-quality SR images from diffusion-based SR models by solving *diffusion ODEs* steadily.

### 3.1 DIFFUSION MODELS, *Diffusion SDEs* AND *Diffusion ODEs*

Diffusion models (Ho et al., 2020; Song et al., 2021b), a kind of generative model, first map samples from an unknown distribution (*e.g.*, the natural image distribution) to samples from a well-known distribution (*e.g.*, the standard Gaussian distribution) by gradually adding noise, and then attempt to revert such process via denoising step by step. The first process is called *forward process*. Taking $\mathbf{x}_0$ as a sample of the unknown distribution $X$, $T$ as the number of noise-adding step, the state $\mathbf{x}_t, t \in [0, T]$ of *forward process* satisfies

$$q(\mathbf{x}_t|\mathbf{x}_0) = \mathcal{N}(\mathbf{x}_t; \alpha(t)\mathbf{x}_0, \sigma^2(t)\mathbf{I}), q(\mathbf{x}_T) = \mathcal{N}(\mathbf{x}_T; \mathbf{0}, \mathbf{I}), \tag{1}$$

where $\alpha(t), \sigma(t)$ are differentiable functions of $t$ defined by hyper-parameters. Furthermore, Kingma et al. (2021) prove that the transition distribution $q(\mathbf{x}_t|\mathbf{x}_0)$ can be given by the following stochastic differential equation (SDE) at any $t \in [0, T]$:

$$\mathrm{d}\mathbf{x}_t = f(t)\mathbf{x}_t\mathrm{d}t + g(t)\mathrm{d}\mathbf{w}_t, \tag{2}$$

where $\mathbf{w}_t$ is a standard Wiener process, and $f(t), g(t)$ are given by

$$f(t) = \frac{\mathrm{d}\log\alpha(t)}{\mathrm{d}t}, g^2(t) = \frac{\mathrm{d}\sigma^2(t)}{\mathrm{d}t} - 2\frac{\mathrm{d}\log\alpha(t)}{\mathrm{d}t}\sigma^2(t). \tag{3}$$

The *reverse process* attempts to learn a parameterized distribution $p_\theta(\mathbf{x}_0)$ to fit the real data distribution $q(\mathbf{x}_0)$ by using a trained noise-prediction model $\boldsymbol{\epsilon}_\theta(\mathbf{x}_t, t)$ to gradually generate $\mathbf{x}_0$ from $\mathbf{x}_T$ (Ho et al., 2020). Lu et al. (2022) prove that the reverse process can be done by solving the following parameterized SDE (*diffusion SDE*) with numerical solvers:

$$\mathrm{d}\mathbf{x}_t = [f(t)\mathbf{x}_t + \frac{g^2(t)}{\sigma(t)}\boldsymbol{\epsilon}_\theta(\mathbf{x}_t, t)]\mathrm{d}t + g(t)\mathrm{d}\bar{\mathbf{w}}_t, \mathbf{x}_T \sim \mathcal{N}(\mathbf{0}, \mathbf{I}), \tag{4}$$

where $\boldsymbol{\epsilon}_\theta(\mathbf{x}_t, t)$ is a trainable noise-prediction neural network and $\bar{\mathbf{w}}_t$ is another standard Wiener process in the reverse time. The original DDPM (Ho et al., 2020) sampler used by current diffusion-based SR models is a discrete-time solver of *diffusion SDE*. When discretizing *diffusion SDEs*, the step sizes are limited because the Wiener process $\bar{\mathbf{w}}_t$ contains randomness. Thus, the resampled DDPM-100 sampler which is mentioned before with larger step sizes performs not satisfying.

Moreover, Song et al. (2021b) give an ordinary differential equation (ODE) which has the same marginal distribution of *diffusion SDE*:

$$\frac{\mathrm{d}\mathbf{x}_t}{\mathrm{d}t} = f(t)\mathbf{x}_t + \frac{g^2(t)}{2\sigma(t)}\boldsymbol{\epsilon}_\theta(\mathbf{x}_t, t), \mathbf{x}_T \sim \mathcal{N}(\mathbf{0}, \mathbf{I}). \tag{5}$$

Such ODE is called *diffusion ODE*. Because *diffusion ODEs* have no randomness, one can get an exact solution $\mathbf{x}_0$ given a BC $\mathbf{x}_T$ by solving the *diffusion ODEs* with corresponding numerical solvers like DDIM (Song et al., 2021a) or DPM-Solver (Lu et al., 2022). Thus, we can use a parameterized projection:

$$\mathbf{x}_0 = h_\theta(\mathbf{x}_T), \mathbf{x}_T \sim \mathcal{N}(\mathbf{0}, \mathbf{I}), \tag{6}$$

to represent the solution of 5. We can extend the diffusion models to conditional ones $p_\theta(\mathbf{x}_0|c)$ by providing conditions $c$ when training the noise-prediction model $\boldsymbol{\epsilon}_\theta(\mathbf{x}_t, c, t)$. By randomly dropping the conditions during the training process, the model can be jointly conditional and unconditional (Ho & Salimans, 2021). We define the projections:

$$\mathbf{x}_0 = h_\theta(\mathbf{x}_T, c), \mathbf{x}_0 = h_\theta(\mathbf{x}_T, \phi), \tag{7}$$

are the solution to conditional *diffusion ODE* and the solution to unconditional *diffusion ODE* of the same diffusion model respectively, where $\phi$ denotes the blank condition which is dropped.

## 3.2 Analyzing Optimal BCs $\mathbf{x}_T^*$ of *Diffusion ODEs* for Diffusion-based SR Models

For image SR tasks, steady SR results mean deterministic samples of the learned conditional distribution $p_\theta(\mathbf{x}_0|c)$, where the conditions $c$ are LR images $\mathbf{y}$. In other words, we should only sample once from the distribution. The parameterized distribution $p_\theta(\mathbf{x}_0|\mathbf{y})$ learned by a well-trained diffusion model is a fitting to the data probability distribution $q(\mathbf{x}_0|\mathbf{y})$ and the training data pairs $(\mathbf{z}_i, \mathbf{y}_i)$ are samples and conditions of the distribution $q(\mathbf{x}_0|\mathbf{y})$, where $\mathbf{z}_i$ denotes the corresponding HR image of $\mathbf{y}_i$. From the perspective of max-likelihood, the $(\mathbf{z}_i, \mathbf{y}_i)$ pairs should be located at the point with the biggest probability distribution $q(\mathbf{x}_0|y_1)$:

$$\mathbf{z}_i = \arg\max_{\mathbf{x}_0} q(\mathbf{x}_0|\mathbf{y}_i). \tag{8}$$

So, the optimal sample of $p_\theta(\mathbf{x}_0|\mathbf{y})$ should satisfy:

$$\mathbf{x}_0^* = \arg\max_{\mathbf{x}_0} p_\theta(\mathbf{x}_0|\mathbf{y}). \tag{9}$$

When we solve *diffusion ODEs* to sample from the diffusion model $p_\theta(\mathbf{x}_0|\mathbf{y})$, we actually sample $\mathbf{x}_T \sim \mathcal{N}(\mathbf{0}, \mathbf{I})$ and project $\mathbf{x}_T$ to final samples $\mathbf{x}_0$ via the projection in Eqn. 7. By leveraging the law of total probability, we can replace the variable from $\mathbf{x}_0$ to $\mathbf{x}_T$, getting the likelihood of $\mathbf{x}_T$:

$$p'_\theta(\mathbf{x}_T|\mathbf{y}) = \sum_{\bar{\mathbf{y}} \in \mathcal{C}} p_\theta(\mathbf{x}_0|\mathbf{y})|_{\mathbf{x}_0 = h_\theta(\mathbf{x}_T, \bar{\mathbf{y}})} p(\bar{\mathbf{y}}) = p_\theta(h_\theta(\mathbf{x}_T, \mathbf{y})), \tag{10}$$

where $\mathcal{C}$ is the theoretically universal set of all LR images and $\bar{\mathbf{y}}$ indicates all the LR images. The proof of Eqn. 10 refers to the Sec. A of the appendix. For unconditional sampling, we have:

$$p'_\theta(\mathbf{x}_T) = p_\theta(\mathbf{x}_0)|_{\mathbf{x}_0 = h_\theta(\mathbf{x}_T, \phi)} = p_\theta(h_\theta(\mathbf{x}_T, \phi)). \tag{11}$$

By substituting Eqn. 10 into Eqn. 9, optimal BCs and samples should satisfy:

$$\mathbf{x}_T^* = \arg\max_{\mathbf{x}_T \sim \mathcal{N}(\mathbf{0}, \mathbf{I})} p_\theta(h_\theta(\mathbf{x}_T, \mathbf{y})), \mathbf{x}_0^* = h_\theta(\mathbf{x}_T^*, \mathbf{y}). \tag{12}$$

Based on Bayesian rule, we have:

$$p_\theta(\mathbf{x}_0|\mathbf{y}) = \frac{p_\theta(\mathbf{x}_0, \mathbf{y})}{p(\mathbf{y})} = \frac{p_\theta(\mathbf{y}|\mathbf{x}_0)}{p(\mathbf{y})} p_\theta(\mathbf{x}_0). \tag{13}$$

By replacing the variable from $\mathbf{x}_0$ to $\mathbf{x}_T$ in Eqn. 13 with Eqn. 10 and Eqn. 11, the parameterized conditional distribution is:

$$p_\theta(h_\theta(\mathbf{x}_T, \mathbf{y})) = \frac{p_\theta(\mathbf{y}|h_\theta(\mathbf{x}_T, \phi))}{p(\mathbf{y})} p_\theta(h_\theta(\mathbf{x}_T, \phi)). \tag{14}$$

In Eqn. 14, $p(\mathbf{y})$ is the prior probability distribution of LR images which is a uniform distribution, and $p_\theta(h_\theta(\mathbf{x}_T, \phi))$ is not related to the LR image $\mathbf{y}$. $p_\theta(\mathbf{y}|h_\theta(\mathbf{x}_T, \phi))$ is an implicit classifier, indicating the probability of image $\mathbf{y}$ is the corresponding LR image of an unconditionally generated image $h_\theta(\mathbf{x}_T, \phi)$. For a well-trained model, such probability is also approximately uniform, because the distribution of unconditionally generated images will be approximate to the real distribution of choosing images in the dataset, which is uniform. Thus, $p_\theta(h_\theta(\mathbf{x}_T, \mathbf{y}))$ is approximately independent to the specific LR images $\mathbf{y}$, which indicates:

$$\mathbf{x}_T^* = \arg\max_{\mathbf{x}_T \sim \mathcal{N}(\mathbf{0}, \mathbf{I})} p_\theta(h_\theta(\mathbf{x}_T, \mathbf{y})) \approx \arg\max_{\mathbf{x}_T \sim \mathcal{N}(\mathbf{0}, \mathbf{I})} p_\theta(h_\theta(\mathbf{x}_T, \mathbf{y}_i)), \forall \mathbf{y}_i \in \mathcal{C}. \tag{15}$$

We design an experiment in the Sec. E of the appendix to validate a derivation of the approximate independence of $p_\theta(h_\theta(\mathbf{x}_T, \mathbf{y}))$ to different $\mathbf{y}$. Hitherto, we have stated that the optimal BC $\mathbf{x}_T^*$ is approximately general for different LR images $\mathbf{y}$. In the next subsection, we depict how to approximate $\mathbf{x}_T^*$ with the criterion of a reference set containing $R$ HR-LR image pairs $\mathcal{R} = \{(\mathbf{z}_i, \mathbf{y}_i)\}_{i=1}^R$ which is a subset of the training dataset.

### 3.3 APPROXIMATING OPTIMAL BCs $\tilde{\mathbf{x}}_T$ OF *Diffusion ODE*s FOR DIFFUSION-BASED SR MODELS

As we have discussed before, a well-trained model $p_\theta(\mathbf{x}_0|\mathbf{y})$ is a fitting of $q(\mathbf{x}_0|\mathbf{y})$. Thus, we can take $q(\mathbf{x}_0|\mathbf{y})$ to substitute $p_\theta(\mathbf{x}_0|\mathbf{y})$ in Eqn. 15, getting an approximation $\tilde{\mathbf{x}}_T$ of $\mathbf{x}_T^*$:

$$\tilde{\mathbf{x}}_T = \underset{\mathbf{x}_T \sim \mathcal{N}(\mathbf{0},\mathbf{I})}{\arg\max} \ q(h_\theta(\mathbf{x}_T, \mathbf{y}_i)). \tag{16}$$

Besides, we have the max-likelihood Eqn. 8 of $q(\mathbf{x}_0|\mathbf{y})$:

$$\mathbf{z}_i = \underset{\mathbf{x}_0}{\arg\max} \ q(\mathbf{x}_0|\mathbf{y}_i) = h_\theta(\underset{\mathbf{x}_T \sim \mathcal{N}(\mathbf{0},\mathbf{I})}{\arg\max} \ q(h_\theta(\mathbf{x}_T, \mathbf{y}_i)), \mathbf{y}_i). \tag{17}$$

Considering the characteristics of natural images, the distribution $q(\mathbf{x}_0|\mathbf{y})$ is a continuous distribution. So, there exists a neighbour around $\mathbf{z}_i$ where $q(\mathbf{x}_0|\mathbf{y}_i)$ is monotonic. Furthermore, the closer $\mathbf{x}_0$ gets to $\mathbf{z}_i$, the bigger $q(\mathbf{x}_0|\mathbf{y}_i)$ is. By taking $M(\cdot, \cdot)$ as the function which measures the distance of two images, the $\tilde{\mathbf{x}}_T$ can be approximated by:

$$\tilde{\mathbf{x}}_T = \underset{\mathbf{x}_T \sim \mathcal{N}(\mathbf{0},\mathbf{I})}{\arg\max} \ q(h_\theta(\mathbf{x}_T, \mathbf{y}_i)) \approx \underset{\mathbf{x}_T \sim \mathcal{N}(\mathbf{0},\mathbf{I})}{\arg\min} \ M(h_\theta(\mathbf{x}_T, \mathbf{y}_i), \mathbf{z}_i). \tag{18}$$

Because the monotonicity of $q(\mathbf{x}_0|\mathbf{y}_i)$ is limited in a small neighbour, we can use a set containing $R$ HR-LR image pairs $\mathcal{R} = \{(\mathbf{z}_i, \mathbf{y}_i)\}_{i=1}^R$ to calculate $\tilde{\mathbf{x}}_T$ to achieve better approximation:

$$\tilde{\mathbf{x}}_T \approx \underset{\mathbf{x}_T \sim \mathcal{N}(\mathbf{0},\mathbf{I})}{\arg\min} \sum_{i=1}^R M(h_\theta(\mathbf{x}_T, \mathbf{y}_i), \mathbf{z}_i). \tag{19}$$

Considering the perceptual characteristics of images, we take LPIPS (Zhang et al., 2018a) as the implementation of $M(\cdot, \cdot)$. Because the projection $h_\theta$ is the solution to *diffusion ODE*, it is difficult to give an analytical result of Eqn. 19. We use the idea of the Monte Carol method to estimate $\tilde{\mathbf{x}}_T$. We randomly sample $K$ $\mathbf{x}_T \sim \mathcal{N}(\mathbf{0},\mathbf{I})$, calculate Eqn. 19 and choose the best one:

$$\tilde{\mathbf{x}}_T \approx \underset{\mathbf{x}_T \in \mathcal{K}}{\arg\min} \sum_{i=1}^R \text{LPIPS}(h_\theta(\mathbf{x}_T, \mathbf{y}_i), \mathbf{z}_i), \tag{20}$$

where $\mathcal{K}$ is the set of randomly sampled $K$ $\mathbf{x}_T \sim \mathcal{N}(\mathbf{0},\mathbf{I})$. Last, given unseen LR images $\mathbf{y}$, the corresponding SR images can be generated by:

$$\tilde{\mathbf{x}}_0 = h_\theta(\tilde{\mathbf{x}}_T, \mathbf{y}). \tag{21}$$

## 4 EXPERIMENTS

In order to demonstrate the effectiveness of the proposed sampling method, we apply our method on two diffusion-based SR models. For bicubic-SR, we train a vanilla model following SR3 (Saharia et al., 2022c) as the baseline. For real-SR, we utilize StableSR (Wang et al., 2023) with $w = 0.5$ without any color fixing and DiffIR (Xia et al., 2023a) as the baseline. It is noted that DiffIR officially employs a 4-step-DDPM sampler without noise which we call "D-4" as shown in Tab. 2.

### 4.1 IMPLEMENTATION DETAILS

**Datasets.** For bicubic-SR, we train the model on the widely-used dataset DF2k (Agustsson & Timofte, 2017; Lim et al., 2017) which containing 3,450 high-resolution images. We train a $64 \times 64 \rightarrow 256 \times 256$ model. The training and architecture details of the bicubic-SR model refer to the Sec. B of the appendix. For real-SR, we directly leverage the official pre-trained model of StableSR (Wang et al., 2023) with $w = 0.5$ without any color fixing.

To test the performances of bicubic-SR, we use 3 different datasets containing DIV2k-test (Agustsson & Timofte, 2017), Urban100 (Huang et al., 2015), B100 (Martin et al., 2001). For DIV2k-test and Urban100, we randomly crop 1,000 $256 \times 256$ patches as HR images and downscale them to $64 \times 64$ patches by bicubic kernel as corresponding LR patches. For B100, we randomly extract 200 patches as the image resolutions in this dataset are not large compared with those in other datasets.

Table 1: Qualitative results of bicubic-SR on test datasets. "$\tilde{\mathbf{x}}_T$" denotes "approximately optimal boundary condition" calculated by the proposed method. The metrics of the bottom 9 rows are all sampled with the same SR3 model trained by us. "DPMS" denotes DPM-Solver (Lu et al., 2022). Numbers of PSNR are calculated on RGB channels. Red numbers denote the best performances and blue numbers denote the second best performances.

| Model (& sampling method) | | DIV2k-test | | Urban100 | | BSD100 | |
|---|---|---|---|---|---|---|---|
| | | LPIPS ↓ | PSNR ↑ | LPIPS ↓ | PSNR ↑ | LPIPS ↓ | PSNR ↑ |
| ESRGAN | | 0.1082 | 28.18 | 0.1226 | 23.04 | 0.1579 | 23.65 |
| RankSRGAN | | 0.1171 | 27.98 | 0.1403 | 23.16 | 0.1714 | 23.80 |
| SRDiff | | 0.1286 | 28.96 | 0.1391 | 23.88 | 0.2046 | 24.17 |
| SR3 | DDPM-1000 | 0.1075 | 28.75 | 0.1165 | 24.33 | 0.1555 | 23.86 |
| | DDPM-250 | 0.1142 | 28.95 | 0.1181 | 24.41 | 0.1621 | 24.00 |
| | DDPM-100 | 0.1257 | 29.16 | 0.1232 | 24.51 | 0.1703 | 24.15 |
| | DPMS-20 | 0.1653 | 27.25 | 0.1413 | 23.46 | 0.2037 | 22.79 |
| | DDIM-50 | 0.1483 | 28.55 | 0.1333 | 24.16 | 0.1823 | 23.75 |
| | DDIM-100 | 0.1571 | 28.16 | 0.1335 | 24.05 | 0.1950 | 23.55 |
| | DPMS-20 + $\tilde{\mathbf{x}}_T$ | 0.1210 | 27.45 | 0.1179 | 23.57 | 0.1687 | 22.81 |
| | DDIM-50 + $\tilde{\mathbf{x}}_T$ | 0.1053 | 28.65 | 0.1164 | 24.26 | 0.1552 | 23.99 |
| | DDIM-100 + $\tilde{\mathbf{x}}_T$ | 0.1032 | 28.48 | 0.1136 | 24.12 | 0.1505 | 23.67 |

Table 2: Qualitative results of real-SR on test datasets. "$\tilde{\mathbf{x}}_T$" denotes "approximately optimal boundary condition" calculated by the proposed method. The metrics of the bottom 3 rows are all sampled with the same StableSR model (Wang et al., 2023). Numbers of PSNR are calculated on RGB channels. Red numbers denote the best performances and blue numbers denote the second best performances.

| Model (& sampling method) | | DIV2k-test | | | RealSR | | |
|---|---|---|---|---|---|---|---|
| | | DISTS ↓ | LPIPS ↓ | PSNR ↑ | DISTS ↓ | LPIPS ↓ | PSNR ↑ |
| RealSR | | 0.3051 | 0.5148 | 22.52 | 0.2532 | 0.3673 | 26.30 |
| BSRGAN | | 0.2253 | 0.3416 | 22.13 | 0.2057 | 0.2582 | 25.52 |
| DASR | | 0.2340 | 0.3444 | 22.02 | 0.2113 | 0.3014 | 26.32 |
| Real-ESRGAN | | 0.2108 | 0.3109 | 22.36 | 0.2020 | 0.2511 | 25.12 |
| KDSR-GAN | | 0.2022 | 0.2840 | 22.92 | 0.2006 | 0.2425 | 26.09 |
| StableSR | DDPM-200 | 0.2010 | 0.3189 | 19.42 | 0.2210 | 0.3065 | 21.37 |
| | DDIM-50 | 0.2217 | 0.3629 | 18.82 | 0.2336 | 0.3536 | 21.24 |
| | DDIM-50 + $\tilde{\mathbf{x}}_T$ | 0.2046 | 0.3169 | 19.55 | 0.2164 | 0.2999 | 22.13 |
| DiffIR | D-4 | 0.1773 | 0.2360 | 22.94 | 0.2076 | 0.2604 | 25.33 |
| | D-4 + $\tilde{\mathbf{x}}_T$ | 0.1772 | 0.2357 | 22.95 | 0.1993 | 0.2419 | 25.82 |

To test the performances of real-SR, we utilize datasets including DIV2k-test (Agustsson & Timofte, 2017) and RealSR (Cai et al., 2019). For DIV2k-test, we employ the ×4 degradation process proposed by Wang et al. (2021), synthesizing 1,000 128×128 LR patches. For RealSR, we randomly crop 1,000 128×128 → 512×512 LR-HR pairs.

**Compared methods and metrics.** This paper proposes a method of sampling from diffusion-based SR models, so, the main baselines are current sampling methods used by other diffusion-based SR models on the same model. For bicubic-SR, we leverage several resampled DDPM samplers and *diffusion ODE* samplers. It is noted that we report the performances of DDPM-1000 (Ho et al., 2020) as upper bounds of previous sampling methods, which serves as evidence of our model's capability. For real-SR, we employ resampled DDPM-200 following the official setting of StableSR (Wang et al., 2023) and DDIM-50 (Song et al., 2021a) as the baseline of *diffusion ODE* solver. We utilize PSNR on RGB channels and LPIPS (Zhang et al., 2018a) as evaluation metrics. For real-SR, we further adopt DISTS (Ding et al., 2020) to demonstrate the generality of the proposed method on diverse perceptual metrics.

Besides, we report the performances of other SOTA SR methods. For bicubic-SR, we show the performances of SRDiff (Li et al., 2022), and GAN-based methods including ESRGAN (Wang et al., 2018) and RankSRGAN (Zhang et al., 2019). For real-SR, we show the performances of RealSR (Ji et al., 2020), BSRGAN (Zhang et al., 2021), Real-ESRGAN (Wang et al., 2021), DASR (Liang et al., 2022), and KDSR-GAN (Xia et al., 2023b). We use the open-source codes and pre-trained models

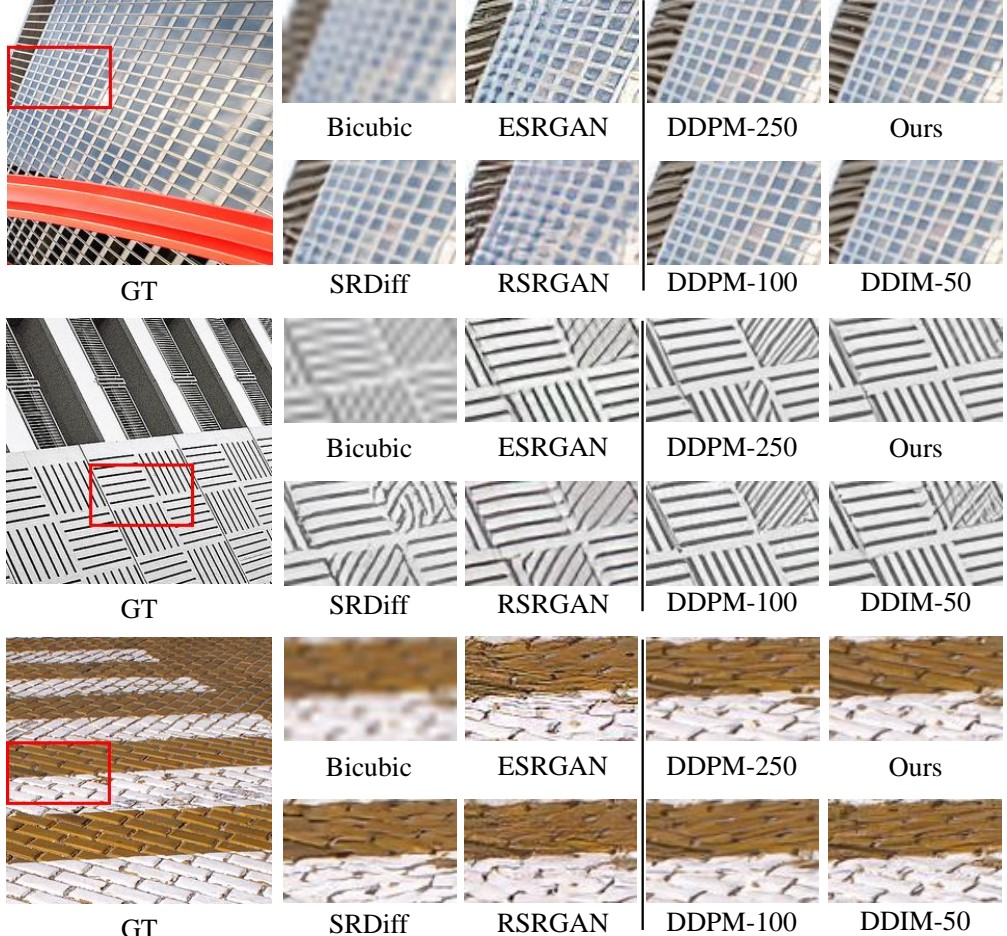

Figure 2: Qualitative comparisons of bicubic-SR results obtained by different methods. "RSRGAN" denotes RankSRGAN (Zhang et al., 2019). All images on the right of the black line are sampled from the same vanilla diffusion-based SR model trained by us. **[Zoom in for best view]**

of these methods without any modification. The details of the source codes of the compared methods refer to the Sec. C of the appendix. To the best of our knowledge, the diffusion-based models with $\tilde{\mathbf{x}}_T$ have achieved superior performances compared to GAN-based SR models. This highlights the effectiveness and efficiency of our method in surpassing the capabilities of GAN-based SR models.

**Settings of calculating $\tilde{\mathbf{x}}_T$ and *diffusion ODE* solvers.** As we have discussed in the Sec. 3.3, we use a reference set $\mathcal{R} = \{(\mathbf{z}_i, \mathbf{y}_i)\}_{i=1}^{R}$ which contains HR-LR image pairs and a set $\mathcal{K}$ which contains $K$ randomly sampled $\mathbf{x}_T$ to calculate the approximately optimal BC $\tilde{\mathbf{x}}_T$. In practice, the $R$ and $K$ are set to 300 and 1,000 respectively for both bicubic-SR and real-SR. The reference sets are synthesized from DIV-2k training set by utilizing the degradation used in the training process. The discussion on the effect of $R$ and $K$ refers to the Sec. 4.3. For *diffusion ODE* solvers, we use DDIM (Song et al., 2021a) on real-SR and further adopt DPM-Solver (Lu et al., 2022) on bicubic-SR to demonstrate that our method can be generally employed to different types of *diffusion ODE* solvers.

## 4.2 QUANTITATIVE AND QUALITATIVE RESULTS

The performances of bicubic-SR and real-SR on testing datasets are shown in Tab. 1 and Tab. 2 respectively. For bicubic-SR, the performance of DDPM-1000 shows the capacity of the model, while the commonly-used sampling methods including DDPM-250, DDPM-100 trade off sample quality for faster sampling speed. It can be seen that the performance of the proposed sampling method with *diffusion ODE* solver of DDIM-100 outperforms all other sampling methods of the same diffusion-based SR model. Remarkably, our method surpasses the previous upper-bound DDPM-1000, which is much slower. For real-SR, our method surpasses the official sampling method of StableSR (Wang

et al., 2023), DDPM-200, with a faster sampling speed, unleashing the capability of StableSR better. Such results demonstrate that we can steadily generate high-quality SR images from the pre-trained diffusion-based SR models by the proposed method. Visual comparisons of bicubic-SR images of different methods are shown in Fig. 2. More visual results can be found in the Sec. F of the appendix.

## 4.3 ABLATION STUDIES

As we have discussed in the Sec. 3.3, we use a reference set $\mathcal{R} = \{(\mathbf{z}_i, \mathbf{y}_i)\}_{i=1}^{R}$ and a set of randomly sampled $\mathbf{x}_T$ $\mathcal{K}$ to estimate the approximately optimal BC $\tilde{\mathbf{x}}_T$. The scales of the two sets will affect the quality of the estimated $\tilde{\mathbf{x}}_T$. The larger $\mathcal{R}$ and $\mathcal{K}$ are, the better estimation of $\tilde{\mathbf{x}}_T$ is. Thus, we perform ablation studies of the scale of the two sets on the task of bicubic-SR.

For the ablation on $\mathcal{R}$, we keep $K = 200$. We build subsets $\mathcal{R}_i$ containing $i$ image pairs and set $i$ to 1, 2, 4, 8, 16. For each $i$, we build 8 $\mathcal{R}_i$ with different random image pairs. With the criterion of each $\mathcal{R}_i$, we choose the corresponding $\tilde{\mathbf{x}}_T$ and test them on a subset of DIV2k test set containing 100 patches with DDIM-50. The mean values and standard deviation values of LPIPS of the SR results with estimated $\tilde{\mathbf{x}}_T$ at each $i$ are shown in Fig. 3.

For the ablation on $\mathcal{K}$, we keep $R = 20$. We randomly sample $i$ $\mathbf{x}_T$ to build sets $\mathcal{K}_i$ and set i to 10, 20, 40, 80, 160. For each $i$, we build 8 $\mathcal{K}_i$ with different $\mathbf{x}_T$. We estimate $\tilde{\mathbf{x}}_T$ from each $\mathcal{K}_i$ and test them on the same subset of DIV2k test set used in the ablation studies on $\mathcal{R}$ with DDIM-50. The mean values and standard deviation values of LPIPS of the SR results with estimated $\tilde{\mathbf{x}}_T$ at each $i$ are also shown in Fig. 3.

It can be seen that the performances become better and steadier as $R$ and $K$ increase.

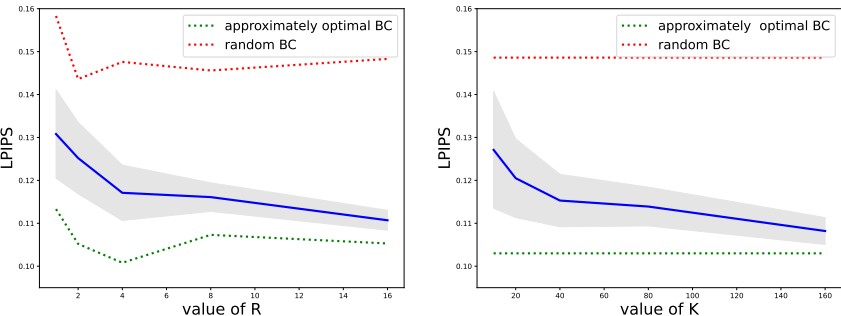

Figure 3: Ablation on values of $R$ and $K$. Shadows denote the standard deviation, the red dotted lines denote LPIPS of SR samples of the subset by DDIM-50 with randomly sampled $\mathbf{x}_T$, indicating the lower-bound of performance, and the green dotted lines denote LPIPS of SR results of the subset by DDIM-50 with $\tilde{\mathbf{x}}_T$, indicating the upper-bound of performance.

## 5 CONCLUSION AND FUTURE WORK

In this work, we propose to steadily sample high-quality SR images from diffusion-based SR models by solving *diffusion ODE*s with approximately optimal BCs $\tilde{\mathbf{x}}_T$. We describe the process of finding these optimal boundary conditions. Experiments show that the proposed sampling method outperforms commonly-used sampling methods for diffusion-based SR models. Our method is not limited to specific architectures of diffusion-based SR models, and does not require additional training. This flexibility allows our method to effectively enhance the sampling performance of pre-trained diffusion-based SR models without any constraints in a plug-and-play manner.

The calculated approximately optimal BC $\tilde{\mathbf{x}}_T$ has the same dimension as LR images $\mathbf{y}$, which can not be directly applied to LR images with other shapes. We will explore designing algorithms which combine the $\tilde{\mathbf{x}}_T$ with resolution-arbitrary sampling methods (Zhang et al., 2023) to achieve the goal of applying our method on LR images with different resolutions. Besides, we only discuss the $\tilde{\mathbf{x}}_T$ in the tasks of image super-resolution. In the future, we will explore further application in other low-level tasks, including image colorization, low-light enhancement, *etc.*

ACKNOWLEDGEMENT

This work was supported in part by the National Natural Science Foundation of China under Grant 62332010, and in part by the Key Laboratory of Science, Technology and Standard in Press Industry (Key Laboratory of Intelligent Press Media Technology).

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

APPENDIX

## A   PROOF TO EQN. 10

**Proposition.** *The likelihood of $\mathbf{x}_T$ which is obtained by replacing the variable from $\mathbf{x}_0$ to $\mathbf{x}_T$ in the likelihood of $\mathbf{x}_0$ satisfies:*

$$p'_\theta(\mathbf{x}_T|\mathbf{y}) = p_\theta(h_\theta(\mathbf{x}_T, \mathbf{y})).$$

**Proof.** First, we apply the law of total probability to all the choices of $\bar{\mathbf{y}} \in \mathcal{C}$, getting:

$$p'_\theta(\mathbf{x}_T|\mathbf{y}) = \sum_{\bar{\mathbf{y}} \in \mathcal{C}} p_\theta(\mathbf{x}_0|\mathbf{y})|_{\mathbf{x}_0=h_\theta(\mathbf{x}_T,\bar{\mathbf{y}})} p(\bar{\mathbf{y}}), \tag{A.1}$$

where $\mathcal{C}$ is the theoretically universal set of all LR images and $\bar{y}$ indicate all the LR images. If $\bar{\mathbf{y}} \neq \mathbf{y}$, $p_\theta(h_\theta(\mathbf{x}_T, \bar{\mathbf{y}})|\mathbf{y})$ would indicate the probability of the generated image $h_\theta(\mathbf{x}_T, \bar{\mathbf{y}})$ being the corresponding SR image of another LR image $\mathbf{y}$, which is almost 0. Thus, $\bar{\mathbf{y}}$ can only be equal to $\mathbf{y}$. Thus, we have:

$$p_\theta(h_\theta(\mathbf{x}_T, \bar{\mathbf{y}})|\mathbf{y}) = \begin{cases} 0, \bar{\mathbf{y}} \neq \mathbf{y} \\ p_\theta(h_\theta(\mathbf{x}_T, \mathbf{y})|\mathbf{y}), \bar{\mathbf{y}} = \mathbf{y} \end{cases}, \tag{A.2}$$

furthermore,

$$\sum_{\bar{\mathbf{y}} \in \mathcal{C}} p_\theta(\mathbf{x}_0|\mathbf{y})|_{\mathbf{x}_0=h_\theta(\mathbf{x}_T,\bar{\mathbf{y}})} p(\bar{\mathbf{y}}) = p_\theta(h_\theta(\mathbf{x}_T, \mathbf{y})|\mathbf{y})p(\mathbf{y}) = p_\theta(h_\theta(\mathbf{x}_T, \mathbf{y}), \mathbf{y}). \tag{A.3}$$

Because $\mathbf{y}$ is the reverse function of $h_\theta(\mathbf{x}_T, \mathbf{y})$ which is deterministic, we have:

$$\sum_{\bar{\mathbf{y}} \in \mathcal{C}} p_\theta(\mathbf{x}_0|\mathbf{y})|_{\mathbf{x}_0=h_\theta(\mathbf{x}_T,\bar{\mathbf{y}})} p(\bar{\mathbf{y}}) = p_\theta(h_\theta(\mathbf{x}_T, \mathbf{y})). \tag{A.4}$$

Hence, the proof is completed. $\qquad\square$

## B   FULL IMPLEMENTATION DETAILS OF THE SR3 BASELINE

The implementation details of the SR3 baseline trained by us contain two parts: details of the diffusion-based SR model $p_\theta(\mathbf{x}_0|\mathbf{y})$ and details of the noise-prediction network used by the diffusion model $\epsilon_\theta(\mathbf{x}_t, \mathbf{y}, t)$. We provide them separately in this section, ensuring the reproducibility of our results.

### B.1   IMPLEMENTATION DETAILS OF THE DIFFUSION MODEL

We use the original diffusion model introduced by Ho et al. (2020) which only predicts the noise in noisy state $\mathbf{x}_t$ without predicting the variances. Thus, the model can be simply trained through the mean square error (MSE) loss between the predicted noise and the real noise. The training loss is:

$$\mathcal{L} = \mathbb{E}_{t,\mathbf{x}_0,\epsilon} ||\epsilon - \epsilon_\theta(\mathbf{x}_t, \mathbf{y}, t)||^2, \tag{B.1}$$

where $\mathbf{y}$ denotes LR images and $\epsilon_\theta(\mathbf{x}_t, \mathbf{y}, t)$ is the noise-prediction network which is particularly depicted in the Sec. B.2. The noise schedule is the same as Ho et al. (2020), which sets $T$ to 1000 and the forward process variances to constants increasing linearly from $\beta_1 = 10^{-4}$ to $\beta_T = 0.02$. During the reverse process of DDPM, we set the variance $\sigma_t$ to $\frac{1-\tilde{\alpha}_{t-1}}{1-\tilde{\alpha}_t}\beta_t$ which performs much better than $\sigma_t = \beta_t$ in resampled few-step sampling, following Nichol & Dhariwal (2021). Following Dhariwal & Nichol (2021), we use a resampled schedule for a few-step sampling. For DDPM-250, we use the schedule of $90, 60, 60, 20, 20$, which is the same as the best schedule for image generation tasks found by Dhariwal & Nichol (2021). For DDPM-100, we use the schedule of $45, 20, 15, 10, 10$, which is not exhaustively swept.

## B.2 Implementation Details of the Noise-Prediction Network

Following most of the current diffusion models (Ho et al., 2020; Saharia et al., 2022c; Li et al., 2022; Shang et al., 2023; Ruan et al., 2023; Ma et al., 2023; Ramesh et al., 2022; Saharia et al., 2022b; Rombach et al., 2022) used in several aspects, we use UNet as the backbone of our noise-prediction network. Following SR3 (Saharia et al., 2022c), the LR images $\mathbf{y}$ are first upsampled by the bicubic kernel to the same size to noise states $\mathbf{x}_t$ and then simply concatenated to noise states $\mathbf{x}_t$ along the channel dimension. The bicubic kernel we used both in downsampling and upsampling is introduced by torchvision (PyTorch-Contributors, 2017) with anti-alias. The architecture of our UNet is similar to the upsampler built by Dhariwal & Nichol (2021) with a small number of parameters. The detailed architecture is shown in Tab. 3. We first train the model for 2M iterations with a batch size of 16, then train the model for another 1M iterations with a batch size of 64, ensuring the convergence of our model. We use Adam optimizer (Kingma & Ba, 2015) during the whole training process and use mixed precision to accelerate training. The total training cost is about 2000 Tesla V100 GPU·hours.

Table 3: Detailed architecture of our UNet used for the diffusion-based SR model.

|  | UNet $64 \rightarrow 256$ |
| --- | --- |
| Model size | 36M |
| Channels | 92 |
| Depth | 2 |
| Channels multiple | 1,1,2,2,3 |
| Heads | 4 |
| Attention resolution | 32,16 |
| BigGAN up/downsample | ✓ |
| Dropout | 0.0 |
| Batch size | $16 \rightarrow 64$ |
| Iterations | 2M + 1M |
| Learning rate | $1e - 4$ |

## C The Sources of the Compared Methods

The sources of the compared methods including SRDiff (Li et al., 2022), ESRGAN (Wang et al., 2018), RankSRGAN (Zhang et al., 2019), RealSR (Ji et al., 2020), BSRGAN (Zhang et al., 2021), and DASR (Liang et al., 2022) are shown in Tab. 4. It is noted that the model of RealSR (Ji et al., 2020) we employ is "DF2K-JPEG".

Table 4: The sources of the compared methods.

| Degradation | Method | URL |
| --- | --- | --- |
| Bicubic | SRDiff | https://github.com/LeiaLi/SRDiff |
|  | ESRGAN | https://github.com/xinntao/ESRGAN |
|  | RankSRGAN | https://github.com/XPixelGroup/RankSRGAN |
| Real | RealSR | https://github.com/jixiaozhong/RealSR |
|  | BSRGAN | https://github.com/cszn/BSRGAN |
|  | DASR | https://github.com/csjliang/DASR |

## D Boosting Mid-Training Models

In the main paper, we analyze all the characteristics of $\mathbf{x}_T^*$ and propose method of approximating $\tilde{\mathbf{x}}_T$ assuming the diffusion-based SR model has been well-trained (*i.e.*, $p_\theta(\mathbf{x}_0|\mathbf{y})$ is a close fit to $q(\mathbf{x}_0|\mathbf{y})$). However, we find that the proposed method can also boost mid-training models. We use the baseline SR3 model of bicubic-SR. The model is trained for only 500k iterations with a batch

Table 5: Performances of the mid-training model with only 500k training iterations. Red numbers denote the best performances among the mid-training model and blue numbers denote the second best performances among the mid-training model.

| Model | Sampling method | DIV2k-test | | Urban100 | | BSD100 | |
|---|---|---|---|---|---|---|---|
| | | LPIPS $\downarrow$ | PSNR $\uparrow$ | LPIPS $\downarrow$ | PSNR $\uparrow$ | LPIPS $\downarrow$ | PSNR $\uparrow$ |
| well-trained | DDPM-1000 | 0.1075 | 28.75 | 0.1165 | 24.33 | 0.1555 | 23.86 |
| | DDPM-250 | 0.1142 | 28.95 | 0.1181 | 24.41 | 0.1621 | 24.00 |
| | DDPM-100 | 0.1257 | 29.16 | 0.1232 | 24.51 | 0.1703 | 24.15 |
| | DDIM-50 | 0.1483 | 28.55 | 0.1333 | 24.16 | 0.1823 | 23.75 |
| | DDIM-50 + $\tilde{\mathbf{x}}_T$ | 0.1053 | 28.65 | 0.1164 | 24.26 | 0.1552 | 23.99 |
| mid-training | DDPM-1000 | 0.2403 | 18.57 | 0.1663 | 19.34 | 0.2269 | 18.77 |
| | DDPM-250 | 0.2361 | 18.65 | 0.1734 | 19.05 | 0.2249 | 18.81 |
| | DDPM-100 | 0.2315 | 18.71 | 0.1640 | 19.30 | 0.2140 | 18.98 |
| | DDIM-50 | 0.4536 | 17.10 | 0.3098 | 17.62 | 0.4040 | 17.84 |
| | DDIM-50 + $\tilde{\mathbf{x}}_T$ | 0.2209 | 19.15 | 0.1618 | 19.97 | 0.2514 | 20.49 |

size of 16, costing 200 Tesla V100 GPU·hours. The performances are shown in Tab. 5. We suspect that the reason for the boosting of the mid-training model is although the mid-training model is not a close fit to $q(\mathbf{x}_0|\mathbf{y})$ yet, it has learned the extreme points of $q(\mathbf{x}_0|\mathbf{y})$. Thus, the assumptions corresponding to extreme points approximately hold (*i.e.*, Eqn. 9, Eqn. 17). So, we still can extract a $\tilde{\mathbf{x}}_T$ based on Eqn. 20 and use it as an approximately optimal BC to other LR images $\mathbf{y}$, getting better performances. We observe that DDIM-50 performs much worse than other sampling methods when applied to the mid-training diffusion-based SR model. Such phenomenon is in conflict with the conclusion of applying these sampling methods in diffusion-based image generation models (Nichol & Dhariwal, 2021). However, our method can still boost the DDIM-50 (*i.e.*, the *diffusion ODE* solver used in the paper) with the approximately optimal BC $\tilde{\mathbf{x}}_T$, reaching comparable performances with DDPM-based sampling methods.

Table 6: Pearson's coefficients between 10 LPIPS sequences of 100 bicubic-SR images for each LR image generated by the baseline SR3 model.

| | LR-1 | LR-2 | LR-3 | LR-4 | LR-5 | LR-6 | LR-7 | LR-8 | LR-9 | LR-10 |
|---|---|---|---|---|---|---|---|---|---|---|
| LR-1 | 1.000 | 0.754 | 0.840 | 0.798 | 0.811 | 0.751 | 0.902 | 0.837 | 0.877 | 0.765 |
| LR-2 | 0.754 | 1.000 | 0.811 | 0.789 | 0.832 | 0.702 | 0.831 | 0.775 | 0.812 | 0.717 |
| LR-3 | 0.840 | 0.811 | 1.000 | 0.732 | 0.799 | 0.654 | 0.841 | 0.799 | 0.836 | 0.745 |
| LR-4 | 0.798 | 0.789 | 0.732 | 1.000 | 0.756 | 0.699 | 0.855 | 0.801 | 0.793 | 0.732 |
| LR-5 | 0.811 | 0.832 | 0.799 | 0.756 | 1.000 | 0.632 | 0.811 | 0.792 | 0.856 | 0.789 |
| LR-6 | 0.751 | 0.702 | 0.654 | 0.699 | 0.632 | 1.000 | 0.721 | 0.734 | 0.611 | 0.704 |
| LR-7 | 0.902 | 0.831 | 0.841 | 0.855 | 0.811 | 0.721 | 1.000 | 0.754 | 0.787 | 0.725 |
| LR-8 | 0.837 | 0.775 | 0.799 | 0.801 | 0.792 | 0.734 | 0.754 | 1.000 | 0.813 | 0.786 |
| LR-9 | 0.877 | 0.812 | 0.836 | 0.793 | 0.856 | 0.611 | 0.787 | 0.813 | 1.000 | 0.801 |
| LR-10 | 0.765 | 0.717 | 0.745 | 0.732 | 0.789 | 0.704 | 0.725 | 0.786 | 0.801 | 1.000 |

# E    VALIDATION ON THE INDEPENDENCE OF $p_\theta(h_\theta(\mathbf{x}_T, \mathbf{y}))$ TO $\mathbf{y}$

As we have stated in the Sec. 3.2, $p_\theta(h_\theta(\mathbf{x}_T, \mathbf{y}))$ is not related to the specific LR images $\mathbf{y}$. In this section, we design an experiment to show the related evidence. As we have mentioned in the Sec. 3.3, we assume distance measurement function $M(h_\theta(\mathbf{x}_T, \mathbf{y}), \mathbf{z})$ has the same shape as $q(h_\theta(\mathbf{x}_T, \mathbf{y}))$ and we use $q(h_\theta(\mathbf{x}_T, \mathbf{y}))$ to approximate $p_\theta(h_\theta(\mathbf{x}_T, \mathbf{y}))$. So, given different LR images $\mathbf{y}_i$, if $p_\theta(h_\theta(\mathbf{x}_T, \mathbf{y}_i))$ are independent, the functions $M(h_\theta(\mathbf{x}_T, \mathbf{y}_i), \mathbf{z}_i)$ of $\mathbf{x}_T$ should have the same shape. Thus, we validate the shapes of $M(h_\theta(\mathbf{x}_T, \mathbf{y}_i), \mathbf{z}_i)$ of different $\mathbf{y}_i$. We randomly sample 10 bicubic-LR image pairs and 100 $\mathbf{x}_T$, then generate 100 SR images by the baseline SR3 model of each LR image and calculate their LPIPS, getting 10 LPIPS sequences. To evaluate the shapes of the 10 LPIPS sequences, we calculate the Pearson correlation coefficients of every two sequences

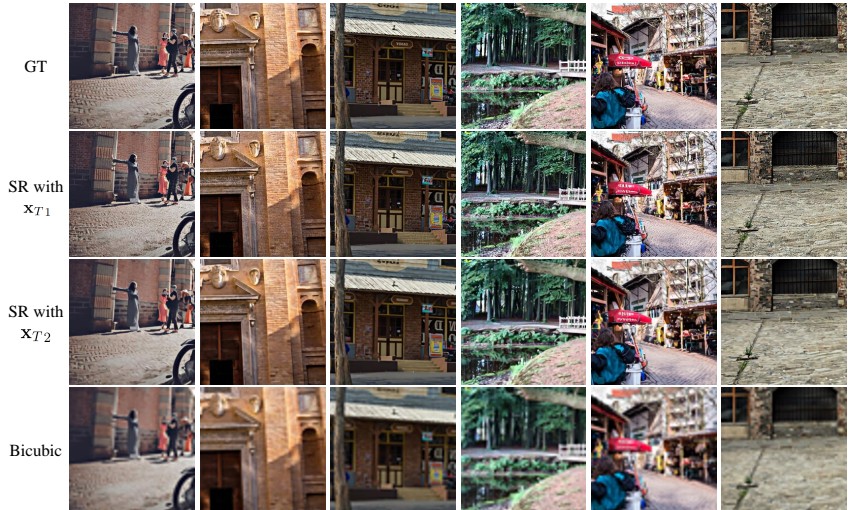

Figure 4: SR results with shared $\mathbf{x}_T$. Results with $\mathbf{x}_{T1}$ all have excessive artifacts and results with $\mathbf{x}_{T2}$ are all over-smooth. Results with shared $\mathbf{x}_T$ share visual features. **[Zoom in for best view]**

and form a matrix shown in Tab. 6. It can be seen that the coefficients are all high, indicating the strong correlation between different LPIPS sequences. To visualize the correlation between SR results of different LR images $\mathbf{y}_i$, we further exhibit several SR images sharing the same $\mathbf{x}_T$ in Fig. 4. It can be seen that SR images of different LR images with the same $\mathbf{x}_T$ have similar visual features. SR results with $\mathbf{x}_{T1}$ seem over-sharp and contain excessive artifacts while SR results with $\mathbf{x}_{T2}$ seem over-smooth. All of them are reasonable but not satisfying enough, indicating the necessity of finding an approximately optimal BC $\tilde{\mathbf{x}}_T$.

It should be noticed that this experiment only validates that the consistency of shapes of $M(h_\theta(\mathbf{x}_T, \mathbf{y}_i), \mathbf{z}_i)$, which is an derivation of the independence of $p_\theta(h_\theta(\mathbf{x}_T, \mathbf{y}))$ to $\mathbf{y}$, instead of the independence itself.

## F    MORE VISUAL RESULTS

In this section, we show more visual results of bicubic-SR compared with ESRGAN (Wang et al., 2018) (which is the representative of GAN-based methods) in Fig. 5, Fig. 6 and Fig. 7, demonstrating the superiority of our method in perceptual quality.

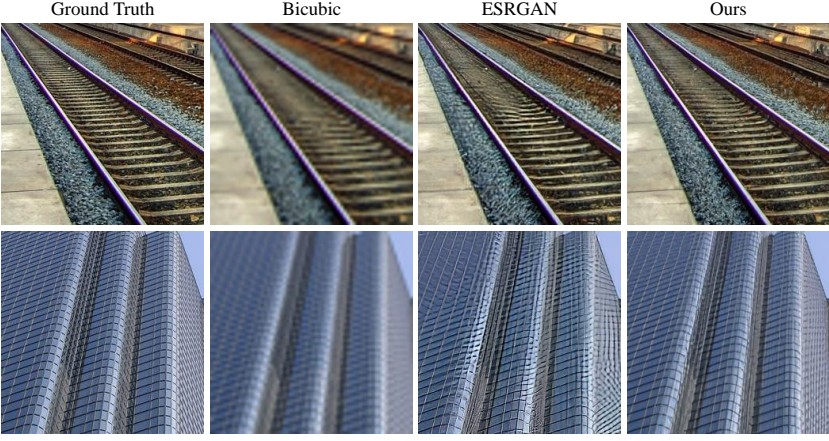

Figure 5: Further visual comparisons. **[Zoom in for best view]**

Ground Truth      Bicubic      ESRGAN      Ours

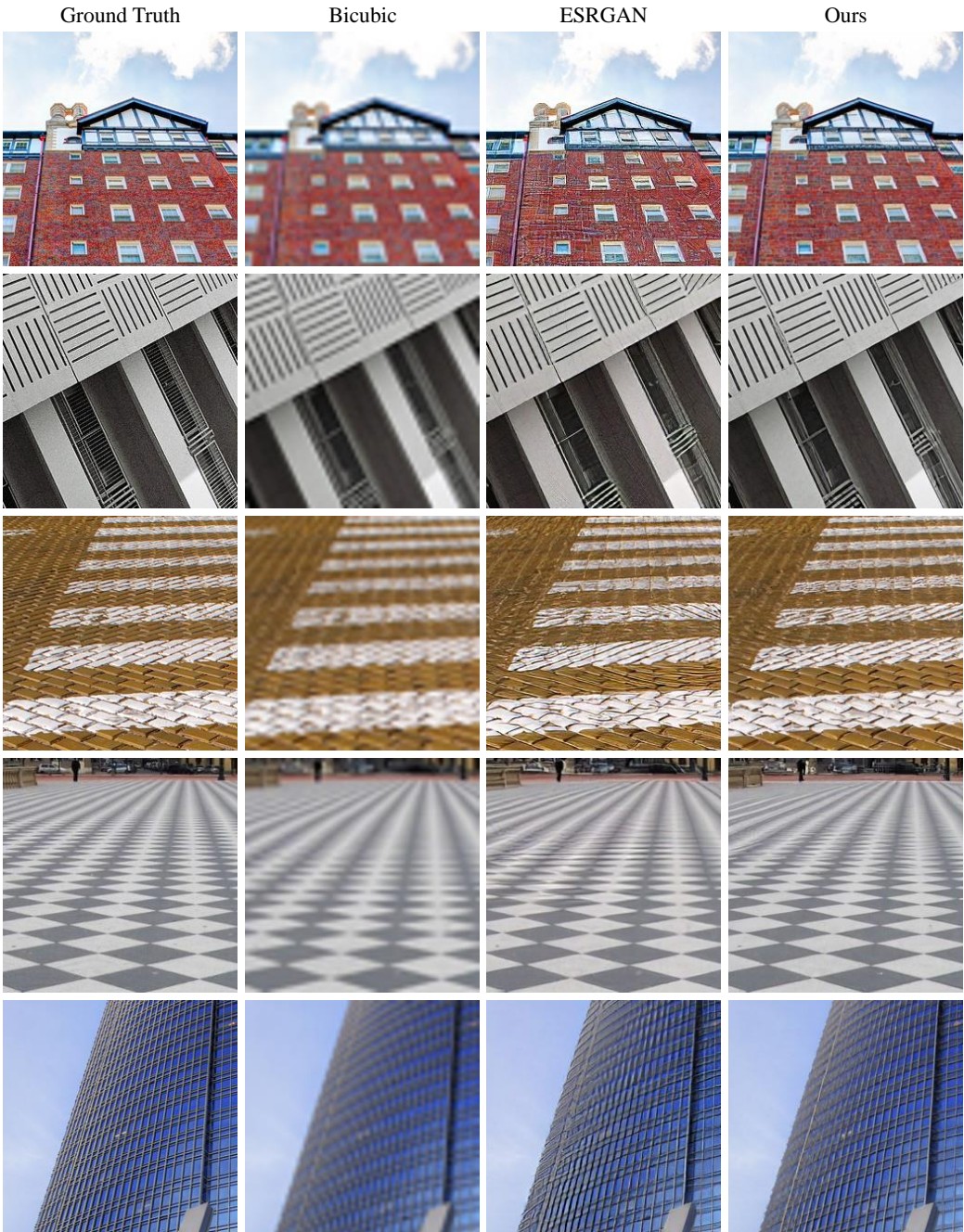

Figure 6: Further visual comparisons. **[Zoom in for best view]**

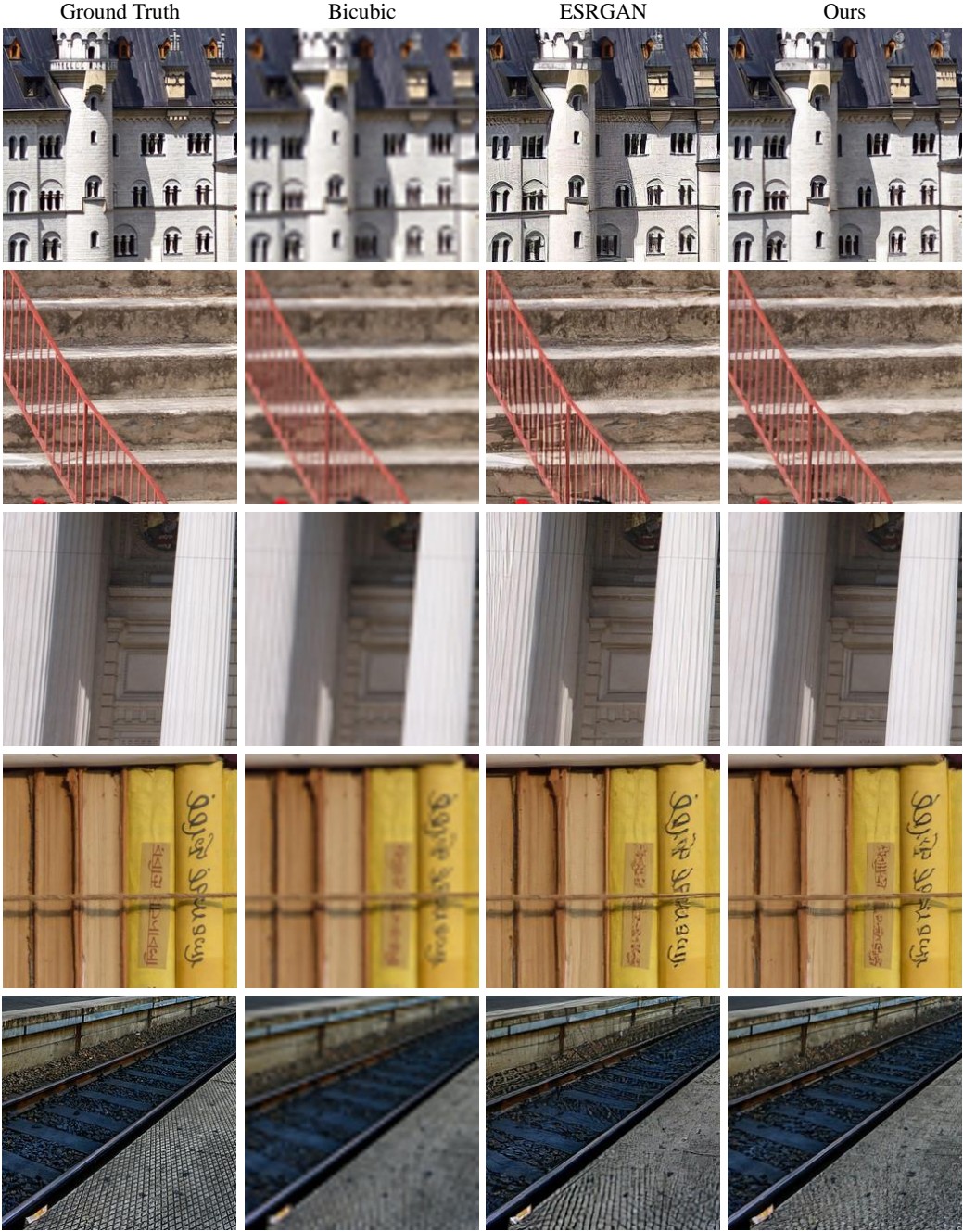

Figure 7: Further visual comparisons. [**Zoom in for best view**]

