# OpenReview forum: "Solving Diffusion ODEs with Optimal Boundary Conditions for Better Image Super-Resolution"
_ICLR.cc/2024/Conference — ICLR 2024 poster_

### Official Review · Reviewer_34bX · 2023-10-17

**Soundness:** 2 fair
**Presentation:** 3 good
**Contribution:** 2 fair
**Rating:** 6
**Confidence:** 4

**Summary:**

This paper propose to steadily sample high-quality SR images from pre-trained diffusion-based SR models by solving diffusion ordinary differential equations with optimal boundary conditions.

**Strengths:**

The paper seems well-written

**Weaknesses:**

1 The methods compared in Table 2 are all outdated. It is necessary to compare them with some state-of-the-art real-world super-resolution tasks [1,2].

2 You need to compare with the state-of-the-art diffusion SR method [3], which also appears to have performed bicubic-SR and real SR tasks.

3 From the figures in the appendix, it seems that the visual improvement is not significant.

4 Please provide a comparison of the computational complexity and runtime for all the methods mentioned in the paper to show your effectiveness.

5 Tables 1 and 2 show that the PSNR is not particularly high, indicating that the network's fidelity is not good. Super-resolution tasks not only seek visual improvement but also place great importance on fidelity. Compared methods have better fidelity. Therefore, I suggest the authors work on improving both PSNR (fidelity) and LPIPS, as this would provide stronger evidence of the effectiveness of your method.


[1] Real-esrgan: Training real-world blind super-resolution with pure synthetic data

[1] Knowledge Distillation based Degradation Estimation for Blind Super-Resolution

[2] Diffir: Efficient diffusion model for image restoration

**Questions:**

see weakness

---

> ### Author Response · Authors · 2023-11-15
> **Authors' responses to Reviewer 34bX (1/2)**
>
> Thank you so much for your time of giving thoughtful review to our paper. About your concerns, we give the responses below. We hope that these responses can solve your questions about our work.
>
> **W1. The methods compared in Table 2 are all outdated. It is necessary to compare them with some state-of-the-art real-world super-resolution tasks [1,2].**
>
> Thanks for the reminder. We have included the comparisons in our rebuttal with more clarifications.
>
> First, the proposed method is a novel sampling method that aims at getting better SR results from existing diffusion-based SR models. Thus, the effectiveness of our method is more revealed by the performance gap of the model with/without our sampling technique, instead of the overall performance comparison, which is demonstrated in existing comparisons (i.e., the bottom 3 rows in Tab. 2).
>
> Second, our comparisons to DASR (published in ECCV 2022) have shown the effectiveness of our method in both LPIPS and NIQE.
>
> Third, we give the results of Real-ESRGAN [1] and KDSR-GAN [2] in the following table.
>
> ||DIV2k-test|||RealSR|||
> |-|-|-|-|-|-|-|
> ||NIQE$\downarrow$|LPIPS$\downarrow$|PSNR$\uparrow$|NIQE$\downarrow$|LPIPS$\downarrow$|PSNR$\uparrow$|
> |Real-ESRGAN|4.805|0.3109|22.36|6.182|0.2511|25.12|
> |KDSR-GAN|5.253|0.2840|22.92|6.673|0.2425|26.09|
> |StableSR + DDPM-200 (official)|4.741|0.3189|19.42|6.185|0.3065|21.37|
> |StableSR + DDIM-50 + $\tilde{\mathbf{x}}_T$ (Ours)|4.309|0.3169|19.55|5.252|0.2999|22.13|
>
> Although our method does not outperform them in LPIPS due to the performance gaps between different baselines, it can still generate SR results with significantly better NIQE.
>
> **W2. You need to compare with the state-of-the-art diffusion SR method [3], which also appears to have performed bicubic-SR and real SR tasks.**
>
> Thanks for the reminder. We will make efforts to add the related results in our updated version.
>
> First, the proposed method is a novel sampling method that aims at getting better SR results from existing diffusion-based SR models, instead of a new diffusion-based SR model. Thus, the vital comparison is between our sampling method and previous sampling methods on the same diffusion-based SR model (i.e., the bottom 3 rows in Tab. 2), as we mentioned in the answer to your question 1.
>
> Second, we leveraged StableSR (https://arxiv.org/abs/2305.07015) as our baseline, which was pre-printed in May 2023. DiffIR [3] (https://arxiv.org/abs/2303.09472) was pre-printed in March 2023. Thus, StableSR is also up-to-date and competitive compared with DiffIR.
>
> Last, we are glad to include the related comparisons. However, in the original paper of DiffIR [3], the real-SR task presented in our paper is not involved (only including image inpainting, bicubic-SR, and motion deblurring). We need more time to train the DiffIR on the task of real-SR to adapt it to our setting and include the related results in the future.
>
> **W3. From the figures in the appendix, it seems that the visual improvement is not significant.**
>
> The visual improvements of SR results are relatively subjective. However, it is still easy to recognize obvious improvements in our visual results in the appendix.
>
> In Fig. 5, our method generates more accurate and clearer crossties of the railway and windows of the building.  In Fig. 6, our method reconstructs more sharp-edged floor tiles and wall bricks with fewer artifacts. In Fig. 7, our method generates more distinct handrails and columns.
>
> In order to make the visual effects more distinctive, we will give residual images to reveal the differences in the camera-ready version. We will also consider conducting a user study to give a quantitative comparison of the results from different methods.

---

> ### Author Response · Authors · 2023-11-15
> **Authors' responses to Reviewer 34bX (2/2)**
>
> **W4. Please provide a comparison of the computational complexity and runtime for all the methods mentioned in the paper to show your effectiveness.**
>
> The comparisons of running time are given in the following tables. All the times are tested on one single RTX 2080Ti GPU.
>
> For bicubic-SR, our baseline is SR3 and resolution of SR images is 256*256. The comparison is:
>
> ||GAN||Diffusion|Diffusion (SR3)|||
> |-|-|-|-|-|-|-|
> |method|ESRGAN|RankSRGAN|SRDiff|DDPM-100 (official)|DDIM-50|DDIM-50 + $\tilde{\mathbf{x}}_T$ (Ours)|
> |Time (Sec/Img)|0.0989|0.2407|5.4523|5.6283|2.9973|2.9968|
>
> For real-SR, our baseline is StableSR and resolution of SR images is 512*512. The comparison is:
>
> ||GAN|||Diffusion (StableSR)|||
> |-|-|-|-|-|-|-|
> |method|DASR|RealSR|BSRGAN|DDPM-200 (official)|DDIM-50|DDIM-50 + $\tilde{\mathbf{x}}_T$ (Ours)|
> |Time (Sec/Img)|0.1058|0.1970|0.1438|22.1815|5.9140|5.9143|
>
> Our method could achieve better performances with fewer sampling steps, which means better efficiency compared with the official sampling methods of existing diffusion-based SR models. The comparison of running time is in proportion to the comparison of sampling steps, which has already been given in the paper. However, because of the characteristics of diffusion models, it is still extremely difficult to achieve shorter running time with diffusion models compared with GAN-based methods which only need one forward process.
>
> **W5. Tables 1 and 2 show that the PSNR is not particularly high, indicating that the network's fidelity is not good. Super-resolution tasks not only seek visual improvement but also place great importance on fidelity. Compared methods have better fidelity. Therefore, I suggest the authors work on improving both PSNR (fidelity) and LPIPS, as this would provide stronger evidence of the effectiveness of your method.**
>
> The authors understand that PSNR is a well-recognized measure for SR. However, in recent years, its drawbacks have been revealed due to its failure to consider sampling randomness in measurement and not robust to simple geometric transformation. Therefore, recent generative model-based SR methods (including GAN-based, flow-based, diffusion-based, and so on) [4, 5, 6, 7] focus on the perceptual quality of SR images, mainly measured by LPIPS. This protocol is also adopted in a recently published paper [8], confirming that worse PSNR but better LPIPS leads to better perceptual quality.
>
> Furthermore, thank you for your valuable suggestion of improving both distortion and perception simultaneously. However, it is extremely challenging to achieve such a goal due to the widely known "Perception-Distortion Tradeoff [9]" which has been studied in depth. We will take your advice into consideration seriously in our future work.
>
> We really appreciate your excellent review of our paper. If our responses do not address all of your concerns, you are welcome to ask us to give more detailed discussions. We are glad to answer your further questions.
>
> [1] Wang, Xintao, et al. "Real-ESRGAN: Training Real-World Blind Super-Resolution with Pure Synthetic Data", ICCVW, 2021.
>
> [2] Xia, Bin, et al. "Knowledge Distillation based Degradation Estimation for Blind Super-Resolution", ICLR, 2022.
>
> [3] Xia, Bin, et al. "DiffIR: Efficient Diffusion Model for Image Restoration", ICCV, 2023.
>
> [4] Ledig, C., et al. "Photo-Realistic Single Image Super-Resolution Using a Generative Adversarial Network", CVPR, 2017.
>
> [5] Wang, Xintao, et al. "ESRGAN: Enhanced Super-Resolution Generative Adversarial Networks", ECCVW, 2018.
>
> [6] Lugmayr, Andreas, et al. "SRFlow: Learning the Super-Resolution Space with Normalizing Flow", ECCV, 2020.
>
> [7] Saharia, Chitwan, et al. "Image Super-Resolution via Iterative Refinement", IEEE TPAMI, 2022.
>
> [8] Gao, Sicheng, et al. "Implicit Diffusion Models for Continuous Super-Resolution", CVPR, 2023.
>
> [9] Blau, Yochai, and Tomer Michaeli. "The Perception-Distortion Tradeoff", CVPR, 2018.

---

> ### Comment · Reviewer_34bX · 2023-11-15
> **Response to author**
>
> Thanks to author's response
>
> However, I need to point out several issues:
>
> 1 The referential significance of NIQE is not substantial. Increasing the proportion of GAN loss easily diminishes it, but it leads to severe artifacts. I am more inclined towards the author's use of FID as an alternative.
>
> 2 DiffIR has reported comparisons on real-world super-resolution in the supplementary material and has open-sourced relevant code and models. You can directly compare them. Please provide information on parameter count, runtime, and performance comparsion.
>
> 3 From a time comparison perspective, it seems that your method takes 30 times longer than GAN methods, but the performance improvement is limited. This is difficult to accept in practical applications.
>
> 4 Both PSNR and LPIPS are crucial. A high LPIPS and low PSNR indicate significant distortion in images, perhaps making them appear better. However, the authenticity of super-resolution results is crucial in many scenarios.
>
> I hope the author can demonstrate the significance of this article to me. Is it effective, fast, or does it achieve high metrics? I am  willing to improve my rating based on such aspects.

---

> > ### Author Response · Authors · 2023-11-16
> > **Further responses to Reviewer 34bX (1/2)**
> >
> > We really appreciate your active participation in the discussion. Your comments help us to provide more robust evidence of the effectiveness of our paper.
> >
> > For your issues, we give further responses below. We hope that they can address your concerns.
> >
> > **I1. The referential significance of NIQE is not substantial. Increasing the proportion of GAN loss easily diminishes it, but it leads to severe artifacts. I am more inclined towards the author's use of FID as an alternative.**
> >
> > Thank you for your reminder to leverage FID to make our evaluation more comprehensive. We provide the comparison between the results of our method and other methods in FID below.
> >
> > ||DIV2k-test|RealSR|
> > |-|-|-|
> > ||FID$\downarrow$|FID$\downarrow$|
> > |Real-ESRGAN|63.8765|139.9400|
> > |KDSR-GAN|55.1053|137.0464|
> > |StableSR + DDIM-50 + $\tilde{\mathbf{x}}_T$ (Ours)|50.1570|130.3912|
> >
> > It can be seen that our method could achieve significantly better FID, demonstrating the superiority of our method.
> >
> > **I2. DiffIR has reported comparisons on real-world super-resolution in the supplementary material and has open-sourced relevant code and models. You can directly compare them. Please provide information on parameter count, runtime, and performance comparison.**
> >
> > Thank you so much for your reminder of DiffIR. We compare with DiffIR on the RealSR test dataset used in our paper (also used in the paper of DiffIR) and further apply the proposed $\tilde{\mathbf{x}}_T$ to DiffIR. The $\tilde{\mathbf{x}}_T$ is extracted with $K=300$ and $R=100$. We denote the official deterministic sampler used by DiffIR (a DDPM sampler without variation) as D-4 (where "D" is the abbreviation of "deterministic"). The results and comparisons are given below.
> >
> > |||RealSR|||
> > |-|-|-|-|-|
> > ||Sample method|FID$\downarrow$|LPIPS$\downarrow$|PSNR$\uparrow$|
> > |StableSR|DDPM-200 (Official)|127.3447|0.3065|21.37|
> > ||DDIM-50|131.6549|0.3536|21.24|
> > ||DDIM-50 + $\tilde{\mathbf{x}}_T$ (Ours)|130.3912|0.2999|22.13|
> > |DiffIR|D-4 (Official)|134.5969|0.2604|25.33|
> > ||D-4 + $\tilde{\mathbf{x}}_T$ (Ours)|124.2071|0.2419|25.82|
> >
> > It can be seen that our method can also be injected into DiffIR and lead to significant performance improvement. Such results further demonstrate the effectiveness of our method.
> >
> > We also give comparisons of running time and number of parameters below.
> >
> > ||Sample method|Running time (Sec/Img)|Number of parameters (M)|
> > |-|-|-|-|
> > |StableSR|DDPM-200 (Official)|22.1815|~890 (same as stable diffusion)|
> > ||DDIM-50 / DDIM-50 + $\tilde{\mathbf{x}}_T$ (Ours)|5.1490|~890|
> > |DiffIR|D-4 (Official) / D-4 + $\tilde{\mathbf{x}}_T$ (Ours)|0.1761|~26|
> >
> > It is noted that our method is a sampling method of current diffusion-based SR models, instead of a new SR model. Thus, our method could be leveraged to sample better SR results from existing diffusion-based models in a plug-and-play manner without any additional parameters.
> >
> > **I3. From a time-comparison perspective, it seems that your method takes 30 times longer than GAN methods, but the performance improvement is limited. This is difficult to accept in practical applications.**
> >
> > As we have mentioned before, our method is a plug-and-play sampling method of existing diffusion-based SR models. Thus, due to the characteristics of diffusion models, it usually takes longer time compared with GAN-based methods.
> >
> > However, along with the development of future diffusion-based SR models, our method is expected to still work. As we have discussed in our paper, our method is not limited to any specific model architecture. Thus, our method is potential to assist the diffusion-based SR models in the future to achieve better performances without any additional parameters and running time, as the experiments on DiffIR we have conducted in the response of I2.

---

> > ### Author Response · Authors · 2023-11-16
> > **Further responses to Reviewer 34bX (2/2)**
> >
> > **I4. Both PSNR and LPIPS are crucial. A high LPIPS and low PSNR indicate significant distortion in images, perhaps making them appear better. However, the authenticity of super-resolution results is crucial in many scenarios.**
> >
> > The authors totally agree that both PSNR and LPIPS are both crucial to the task of SR. However, due to the performance gap between our baseline methods and other methods, there is still performance improvement space in the PSNRs of our method.
> >
> > The experiments on DiffIR we have shown in the response of I2 reveal the potential of our method. Due to the excellent design of DiffIR which can be used as a baseline, the performances on both LPIPS and PSNR of our method can be more promising.
> >
> > **I5. I hope the author can demonstrate the significance of this article to me. Is it effective, fast, or does it achieve high metrics?**
> >
> > We express the significance of our work in a progressive way.
> >
> > First, the proposed method is a high-quality sampling method with higher speed. Because the proposed method does not have any limitation in model architectures, it can be used in a plug-and-play manner by different diffusion-based SR models to assist them to get better performances without additional parameters.
> >
> > Second, as we have stated in our paper, our method does not have any limitation to the degradation model. This indicates that our method can be used in a wide variety of low-level vision tasks (including low-light enhancement, de-raining, and so on). We will do more exploration in the future.
> >
> > Last, from the perspective of image generation, the proposed $\tilde{\mathbf{x}}_T$ can be utilized to generate high-quality images compared with other randomly sampled $\mathbf{x}_T$. This implies the proposed $\tilde{\mathbf{x}}_T$ "encodes" the characteristics of "high-quality images" by some means. We will manage to give more corresponding analysis on this topic in more general image-generation tasks. Such a phenomenon would have some inspiration to other researchers who are also studying diffusion models from the more general generation perspective.
> >
> > Thank you again for your participation of discussion. If you have any further questions, we will be so glad to give responses to you.

---

> > > ### Comment · Reviewer_34bX · 2023-11-16
> > > **Response to authors**
> > >
> > > Thanks for your response.
> > >
> > > 1. The author should revise their paper by presenting results on the best-performing DiffIR and comparing them with SOTA RealSR methods, such as real-esrgan and KDSR in order to demonstrate their real contribution to pushing the upper bounds in the field.
> > >
> > > 2. I want to know whether the author will release their codes and models for public application.

---

> ### Author Response · Authors · 2023-11-16
> **Further responses to Reviewer 34bX**
>
> Thank you so much for your reply. We respond to your concerns below.
>
> **Q1. The author should revise their paper by presenting results on the best-performing DiffIR and comparing them with SOTA RealSR methods, such as real-esrgan and KDSR in order to demonstrate their real contribution to pushing the upper bounds in the field.**
>
> Thank you so much for your suggestion. We will add the results on DiffIR in the next version of our paper to demonstrate the superiority of the proposed method.
>
> **Q2. I want to know whether the author will release their codes and models for public application.**
>
> Our method employed several codebases because of the different tasks.
>
> For bicubic-SR, we will release the code (including the diffusion model itself and the code of extracting the $\tilde{\mathbf{x}}_T$), the re-implemented SR3 model, and the $\tilde{\mathbf{x}}_T$ used in the corresponding experiments if our paper is accpted.
>
> For StableSR and DiffIR, we directly leveraged their open-source codes (StableSR: https://github.com/IceClear/StableSR, DiffIR: https://github.com/Zj-BinXia/DiffIR) and the corresponding models. We really appreciate their contributions to the community. We will release the $\tilde{\mathbf{x}}_T$ used in the corresponding experiments and the codes of extracting the $\tilde{\mathbf{x}}_T$ if our paper is accpted.
>
> Thank you again for your time in giving reviews and suggestions for our paper. All of your comments really assist us in demonstrating the superiority of our paper and make our paper more robust.

---

> > ### Comment · Reviewer_34bX · 2023-11-16
> > **Response to author**
> >
> > The author addressed some of my concerns. I decide to improve my rating

---

> > > ### Author Response · Authors · 2023-11-16
> > > **Thank you for your appreciation**
> > >
> > > Thank you so much for your reviews, suggestions, and appreciation of our work. We will take all of your comments into consideration seriously and revise our paper.

---

### Official Review · Reviewer_8EEt · 2023-10-30

**Soundness:** 4 excellent
**Presentation:** 3 good
**Contribution:** 4 excellent
**Rating:** 8
**Confidence:** 5

**Summary:**

The paper analyzes the characters of boundary conditions (BCs) in the diffusion-ODE sampling process of diffusion-based Super-Resolution (SR) models and finds out that the optimal BC is shared by different LR images approximately. Based on the analysis, the paper further proposes a method of obtaining an approximately optimal BC based on a reference LR-HR set. The derivation of the paper is mathematically complete. Experiments on the tasks of both bicubic-SR and real-SR demonstrate the superiority of the proposed approximately optimal BC.

**Strengths:**

1. The motivation of analyzing the BCs of diffusion ODEs is intuitive. Different BCs would apparently lead to different SR results. It is vital to find the rule of how BCs affect the results and to propose a method to get a better BC.
2. The analysis of the paper is mathematically complete. The paper proposes the concept of optimal BC and proves that such an optimal BC is common to different LR images. The conclusion is seemingly solid.
3. The experiments are sufficient enough to support the theory. The paper claims that the method of obtaining the approximately optimal BS is not related to the degradation model. Experiments on the tasks of bicubic-SR and real-SR demonstrate the assertion. Further ablation studies show the influence of reference set and the set of BCs.

**Weaknesses:**

1. The proposed method leverages LPIPS as the implementation of distance measurement function M(·,·). Can we leverage pixel-level metrics like negative PSNR as M? The authors should give more discussions.
2. The paper assumes that the model is well-trained. However, even a “well-trained” model cannot fit the real data distribution precisely. Thus, does such an assumption cause potential inaccuracy of the conclusion?
3. The paper claims that p_θ (y|h_θ (x_T,ϕ)) is approximately uniform. However, what ensures that the model does not have biases when leveraging the “blank token”, which is essentially a placeholder token (that doesn't actually exist)?
4. It seems that the process of calculating the approximately optimal BC is time-consuming. How long does it take?

**Questions:**

1. The paper only discusses the context of SR (and other low-level tasks in the Sec. 5). But it seems that the theory does not limit the relationship between guidance and generated results. Do the authors think that the method can be leveraged in more general generation tasks such like text-to-image generation?
2. It seems that the improvement on StableSR (in the task of real-SR) is less than the improvement on SR3 (in the task of bicubic-SR). Could the authors discuss the reasons for this phenomenon?

---

> ### Author Response · Authors · 2023-11-15
> **Authors' responses to Reviewer 8EEt (1/2)**
>
> Thank you so much for your thoughtful review which helps us to improve the quality of our paper. For your questions, we have the following responses. We hope that they can help to address your concerns. Sincerely looking forward to your reply.
>
> **W1. The proposed method leverages LPIPS as the implementation of distance measurement function $M(·,·)$. Can we leverage pixel-level metrics like negative PSNR as $M$? The authors should give more discussions.**
>
> We can also use MSE (which is equivalent to negative PSNR) as the implementation and give results on bicubic-SR in the following table.
> ||DIV2k-test||Urban100||BSD100||
> |-|-|-|-|-|-|-|
> ||LPIPS|PSNR|LPIPS|PSNR|LPIPS|PSNR|
> |DDIM-50 + LPIPS-oriented $\tilde{\mathbf{x}}_T$|0.1053|28.65|0.1164|24.26|0.1552|23.99|
> |DDIM-50 + MSE-oriented $\tilde{\mathbf{x}}_T$|0.1060|28.56|0.1164|24.24|0.1557|24.01|
>
> The results are slightly worse than the LPIPS-oriented $\tilde{\mathbf{x}}_T$ which is used in our paper. This is because MSE is not a less optimal metric to evaluate such distance. Lower MSE may not mean better image quality, which is a consensus in the domain of image SR [1, 2].
>
> **W2. The paper assumes that the model is well-trained. However, even a “well-trained” model cannot fit the real data distribution precisely. Thus, does such an assumption cause potential inaccuracy of the conclusion?**
>
> Indeed, the gap between the parameterized distribution of diffusion models and real data distribution will affect the accuracy of our analysis, and this is the reason that our $\tilde{\mathbf{x}}_T$ is an "approximately" optimal boundary condition. We have pointed it out in our paper (in the text description before the Eqn. 16). In order to mitigate the inaccuracy, we leverage a reference set of LR-HR image pairs to find the $\tilde{\mathbf{x}}_T$, instead of one single LR-HR pair.
>
> **W3. The paper claims that $p_\theta (\mathbf{y}|h_\theta (\mathbf{x}_T, \phi))$ is approximately uniform. However, what ensures that the model does not have biases when leveraging the “blank token”, which is essentially a placeholder token (that doesn't actually exist)?**
>
> As you mentioned, the blank token $\phi$ is only symbolic guidance, which means not give any guidance to the model when generating. When generating images from $\phi$, we are actually unconditionally generating images (such unconditional generating can be achieved by randomly dropping the LR images during training [3]). For $p(\mathbf{x}_0|\phi)$, a "well-trained" model means it should be close to the training data distribution $q(\mathbf{x}_0|\phi)$, which is uniform.
>
> Furthermore, we give more discussion on our claim, "for a well-trained model, such probability (the probability of unconditionally generating an image which is the corresponding SR image of the LR image y) is also approximately uniform". For a "well-trained" model, such probability is close to the probability of "randomly sampling an image which is the corresponding SR image of a certain LR image $\mathbf{y}$ from the HR image dataset". The distribution of the latter one is consistent with sampling training data during the training process, which is uniform for different $\mathbf{y}$. Thus, such probability will also be uniform.
>
> **W4. It seems that the process of calculating the approximately optimal BC is time-consuming. How long does it take?**
>
> The time of calculating the approximately optimal BC is about 20 hours on a V100 GPU. As we have mentioned in our paper, we take $R=300$ and $K=1000$ when calculating the approximately optimal BC. In practice, we did not simply sample $R \cdot K=300,000$ SR images. We first test all the $K=1000$ $\mathbf{x}_T$ on a small reference set containing 10 LR-HR pairs and only retain the top 300 $\mathbf{x}_T$ as a pre-screening process. Then we leverage a larger reference set and repeat the filtering process. At last, the final reference set contains all the 300 reference pairs.
>
> Such a time cost is a one-off. We just need to calculate the approximately optimal BC once for one diffusion-based SR model. This time cost is much lower than the time cost of training the model itself. Thus, such a process is not time-consuming.

---

> ### Author Response · Authors · 2023-11-15
> **Authors' responses to Reviewer 8EEt (2/2)**
>
> **Q1. The paper only discusses the context of SR (and other low-level tasks in the Sec. 5). But it seems that the theory does not limit the relationship between guidance and generated results. Do the authors think that the method can be leveraged in more general generation tasks such like text-to-image generation?**
>
> Your suggestion of leveraging the method in other tasks is so valuable. As we have mentioned in our paper, the proposed method can also be used in other low-level vision tasks. However, maybe it is not suitable enough for high-level generation tasks. Because these tasks do not have unique ground truths, it is difficult to define the concept of "optimal sample" which was defined by the ground truth (to be more specific, HR images in our paper). For example, in the task of text-to-image generation, as you mentioned, there can be several images that are semantically aligned to the input text. In other words, there is no exact "ground truth" to an input text.
>
> Thank you so much for your suggestion. We will consider your advice seriously in our future works of leveraging our method in more low-level vision tasks.
>
> **Q2. It seems that the improvement on StableSR (in the task of real-SR) is less than the improvement on SR3 (in the task of bicubic-SR). Could the authors discuss the reasons for this phenomenon?**
>
> We think that the reason to the performance gap on StableSR and SR3 is that StableSR builds diffusion model in the latent space, while SR3 builds diffusion model in the image space. However, we actually propose the concept of "optimal sample" in the image space (Eqn. 9). Thus, the diffusion process in the latent space affects the accuracy of the approximation of our method (Eqn. 15, 16, 18) slightly, causing less improvement of StableSR. However, our method still helps StableSR to sample better results than its official sampling method (DDPM-100) in shorter time.
>
> Thank you again for your approval of our paper and your valuable review. You are welcome to ask us to give further responses to address your concerns.
>
> [1] Ledig, C., et al. "Photo-Realistic Single Image Super-Resolution Using a Generative Adversarial Network", CVPR, 2017.
>
> [2] Wang, X., et al. "ESRGAN: Enhanced Super-Resolution Generative Adversarial Networks", ECCVW, 2018.
>
> [3] Ho, J., and Salimans, T.. "Classifier-Free Diffusion Guidance", NIPSW, 2021.

---

### Official Review · Reviewer_4gay · 2023-10-31

**Soundness:** 2 fair
**Presentation:** 3 good
**Contribution:** 2 fair
**Rating:** 6
**Confidence:** 3

**Summary:**

This paper focuses on the randomness in the inverse process of the diffusion model applied to super-resolution tasks, which makes it difficult to ensure the quality of SR results. By solving diffusion ordinary differential equations with optimal boundary conditions, the authors propose an efficient plug-and-play method that enables diffusion models to stably sample high-quality SR images with fewer sampling steps.

Post rebuttal:
I have read the rebuttal and would like to raise my score a little bit.

**Strengths:**

1.	The proposed method achieves good visualization results with fewer steps in SR tasks.
2.	The proposed method has good flexibility and can be applied to multiple diffusion models

**Weaknesses:**

1.	The experiments are insufficient. Although the proposed method has achieved good results on LPIPS, the PSNR values on multiple test sets are very low. The author did not discuss it in depth and did not show the results of SSIM.
2.	The paper stated that the proposed method has fewer parameters and is more efficient than the GAN method, but did not show the corresponding comparison results. Such as comparison of specific parameters or running time.

**Questions:**

Further supplement and improve the experiment, please refer to the Weaknesses for details.

---

> ### Author Response · Authors · 2023-11-15
> **Authors' responses to Reviewer 4gay**
>
> Thank you so much for your reviews and comments of our paper. Your summary of our paper is precise and concise. About your concerns, we give responses below. We hope they can address your concerns.
>
> **W1. The experiments are insufficient. Although the proposed method has achieved good results on LPIPS, the PSNR values on multiple test sets are very low. The author did not discuss it in depth and did not show the results of SSIM.**
>
> First, the authors understand that PSNR and SSIM are well-recognized measures for SR. However, in recent years, their drawbacks have been revealed due to their failure to consider sampling randomness in measurement and not robust to simple geometric transformation. Therefore, recent generative model-based SR methods (including GAN-based, flow-based, diffusion-based, and so on) [1, 2, 3, 4] focus on the perceptual quality of SR images, mainly measured by LPIPS. This protocol is also adopted in a recently published paper [5], confirming that worse PSNR but better LPIPS can lead to better perceptual quality.
>
> Second, we appreciate the reviewer's comments. The evaluations of SR are still open questions. We will thoroughly discuss and address these issues in our revised paper.
>
> **W2. The paper stated that the proposed method has fewer parameters and is more efficient than the GAN method, but did not show the corresponding comparison results. Such as comparison of specific parameters or running time.**
>
> Sorry for the confusion. We make the further clarification as follows.
>
> First, our method is a sampling method of current diffusion-based SR models, instead of a new SR model. Thus, our method could be leveraged to sample better SR results from existing methods without any additional parameters.
>
> Second, our method could achieve better performances with fewer sampling steps, which means better efficiency compared with the baseline diffusion-based SR model. The comparison of running time is in proportion to the comparison of sampling steps, which is already given in the paper. However, because of the characteristics of diffusion models, it is still extremely difficult to achieve faster running time compared with GAN-based methods which only need one forward process. The comparisons of running time among different sampling methods from one diffusion-based SR model are given in the following tables. All the times are tested on one single RTX 2080Ti GPU.
>
> For bicubic-SR, the comparison on baseline SR3 is (resolution 256*256):
> ||DDPM-1000|DDPM-250|DDPM-100 (SR3 official)|DDIM-50|DDIM-50 + $\tilde{\mathbf{x}}_T$ (Ours)|
> |-|-|-|-|-|-|
> |Time (Sec/Img)|54.3940|13.6769|5.6283|2.9973|2.9968|
>
> For real-SR, the comparison on baseline StableSR is (resolution 512*512):
>
> ||DDPM-200 (StableSR official)|DDIM-50|DDIM-50 + $\tilde{\mathbf{x}}_T$ (Ours)|
> |-|-|-|-|
> |Time (Sec/Img)|22.1815|5.9140|5.9143|
>
> Our statement refers to that our method is able to sample better SR results compared with GAN-based methods from current diffusion-based SR models, achieving better efficiency and effectiveness. Such diffusion-based SR models cannot outperform GAN-based methods with previous sampling methods. We apologize if our statement leads to ambiguity. We will refine the corresponding statement in the next edition of our paper to make it more precise.
>
> Thank you again for your review of our paper. You are welcome to further ask us about your concerns which our responses do not fully address and ask us to give more detailed responses. Really looking forward to your reply.
>
> [1] Ledig, C., et al. "Photo-Realistic Single Image Super-Resolution Using a Generative Adversarial Network", CVPR, 2017.
>
> [2] Wang, Xintao, et al. "ESRGAN: Enhanced Super-Resolution Generative Adversarial Networks", ECCVW, 2018.
>
> [3] Lugmayr, Andreas, et al. "SRFlow: Learning the Super-Resolution Space with Normalizing Flow", ECCV, 2020.
>
> [4] Saharia, Chitwan, et al. "Image Super-Resolution via Iterative Refinement", IEEE TPAMI, 2022.
>
> [5] Gao, Sicheng, et al. "Implicit Diffusion Models for Continuous Super-Resolution", CVPR, 2023.

---

### Comment · Area_Chair_n4ou · 2023-11-23
**[ICLR 2024 Reviewers’ feedback] Please read authors’ responses and give your feedback**

Dear Reviewers,

Thanks again for your strong support and contribution as an ICLR 2024 reviewer.

Please check the response and other reviewers’ comments. You are encouraged to give authors your feedback after reading their responses. Thanks again for your help!

Best,

AC

---

### Meta-Review · Area_Chair_n4ou · 2023-12-13

**Metareview:**

The paper is well-written. The motivation for analyzing the BCs of diffusion ODEs is intuitive. The analysis of the paper is mathematically complete. The proposed method achieves good visual results with fewer steps in image SR. But, more experiments still need to be added to the revision. For example, as one reviewer points out, more recent methods (e.g., DiffIR) should be compared. After rebuttal and extensive discussions, all reviewers gave positive scores.

**Justification For Why Not Higher Score:**

More experiments still need to be added to the revision. For example, as one reviewer points out, more recent methods (e.g., DiffIR) should be compared.

**Justification For Why Not Lower Score:**

After rebuttal and extensive discussions, all reviewers gave positive scores.

---

### Decision · Program_Chairs · 2024-01-16

Accept (poster)